# Hsp70 is phosphorylated in a conserved response to DNA damage and contributes to cell cycle control

**Thomas Moss[1,2], Alexandra Wooldredge[1,2], Koustav Bhakta[1,2], Matthew Cronin[1,2], Jason E Gestwicki[3], Shaeri Mukherjee[1,2,4]***

[1]G.W. Hooper Foundation, University of California, San Francisco, San Francisco, United States; [2]Department of Microbiology and Immunology, University of California, San Francisco, San Francisco, United States; [3]Department of Pharmaceutical Chemistry and the Institute for Neurodegenerative Diseases, University of California, San Francisco, San Francisco, United States; [4]Biohub, San Francisco, United States

## eLife Assessment

This potentially **valuable** manuscript focuses on the phosphorylation of residue T495 as a mechanism to inactivate HSP70 and disrupt cell cycle progression in response to DNA damage. The evidence supporting this model is **solid**, but would be significantly strengthened by additional studies defining the extent of T495 phosphorylation induced by DNA damage, identifying the kinase responsible for phosphorylating T495 of HSP70, and further elucidation of the functional implications of T495 phosphorylation in human cells. This work will be of interest to scientists focused on topics including chaperone biology, proteostasis, cell cycle progression, and DNA damage.

***For correspondence:**
Shaeri.Mukherjee@ucsf.edu

**Abstract** Hsp70s are essential molecular chaperones that are increasingly recognized to be regulated by post-translational modifications. Here, we show that phosphorylation of a conserved threonine (T495), previously shown to be exploited by a *Legionella pneumophila* kinase to inhibit Hsp70, occurs endogenously in human cells in response to DNA damage, particularly when base excision repair is overburdened. This modification is cell cycle dependent, and in yeast, phosphomimetic or phosphonull Hsp70 variants disrupt G1/S progression under normal and DNA-damaging conditions. Biochemically, the phosphomimetic T495E mutation locks Hsp70 in an open-like conformation without blocking substrate engagement. Together, our results reveal a conserved mechanism by which dynamic Hsp70 phosphorylation regulates the G1/S transition and delays cell cycle progression during DNA damage, highlighting how pathogen-derived insights can uncover fundamental cell biology principles.

## Introduction

Hsp70s are highly conserved molecular chaperones with wide-ranging roles in cellular homeostasis. At the core of their function is an ATP-driven conformational cycle: Hsp70s alternate between an open ATP-bound state and a closed ADP-bound state, enabling them to bind and release client proteins in a nucleotide-dependent manner (*Rosenzweig et al., 2019*). Through this mechanism, Hsp70s participate in diverse biological processes, including protein homeostasis (*Rosenzweig et al., 2019*), metabolic regulation (*Kaushik and Cuervo, 2018*), the DNA damage response (DDR) (*Dubrez et al., 2020*), and cell cycle control (*Truman et al., 2012*; *Chen et al., 2014*; *Vergés et al., 2007*).

Beyond their central roles in cell biology, Hsp70s have been implicated in the pathology of numerous diseases. For example, Hsp70s can inhibit amyloid-beta aggregation in vitro and reduce its accumulation in mouse neurons, suggesting protective roles in neurodegenerative diseases such as Alzheimer's disease (*Evans et al., 2006*; *Valle-Medina et al., 2025*). Hsp70s are also overexpressed in various cancers and influence tumor cell survival and proliferation (*Calderwood and Murshid, 2017*). Thus, understanding how Hsp70 activity is regulated has broad implications for both basic biology and therapeutic development.

Hsp70 function is tightly controlled by co-chaperones and post-translational modifications (PTMs). Cochaperones – namely J proteins and nucleotide exchange factors – catalyze nucleotide hydrolysis and exchange, facilitate and specify client engagement, and determine client fate (*Rosenzweig et al., 2019*). In recent years, PTMs have emerged as an important layer of regulation (*Truman et al., 2012*; *Ham et al., 2014*; *Preissler et al., 2015*; *Wang et al., 2014*; *Seo et al., 2016*; *Cho et al., 2012*). A computational analysis of the budding yeast Hsp70, Ssa1, identified two conserved 'hotspots' in the nucleotide-binding domain (NBD) and substrate-binding domain (SBD), which are likely regions of PTM-mediated regulation (*Beltrao et al., 2012*). Experimental work has validated some of the PTMs at these sites. For example, T36 in the NBD of Ssa1 is phosphorylated both in response to nutrient stress and exposure to mating pheromone (α-factor), impairing chaperone activity and leading to cell cycle arrest (*Truman et al., 2012*). Our lab showed that the *Legionella pneumophila* (*L.p.*) kinase LegK4 phosphorylates cytosolic human Hsp70 (Hsc70; HSPA8) at T495 during infection, increasing its association with polysomes and globally reducing protein synthesis (*Moss et al., 2019*). The endoplasmic reticulum (ER) resident Hsp70, BiP, is AMPylated at the analogous site (T518) in both mammalian and insect cells, locking it in an 'open' conformation and tuning its activity to the unfolded protein burden in the ER (*Preissler et al., 2015*; *Preissler et al., 2017b*; *Casey et al., 2017*). This indicates that both AMPylation and phosphorylation at this site have regulatory consequences. These findings led us to ask whether phosphorylation at T495 in cytosolic Hsp70 occurs outside of *L.p.* infection and whether it serves endogenous regulatory functions. Mining phosphoproteomics datasets revealed that the homologous residue in *Saccharomyces cerevisiae* Ssa1, T492, is phosphorylated during DNA alkylation damage and when mitotic exit is prevented (*Albuquerque et al., 2008*; *Holt et al., 2009*). However, the functional consequences of this modification have not been studied. In this work, we show that Hsp70 is phosphorylated at T495 (pHsp70) in human cells during base excision repair (BER) of DNA damage. We find that a phosphomimetic mutant T495E allosterically impairs ATP hydrolysis in vitro and stabilizes an open-like conformation while still permitting substrate engagement. In yeast, the analogous mutation T492E causes a growth defect, leads to accumulation of cells in G1, and delays cell cycle re-entry after alkylation damage. Likewise, the phosphonull mutation (T492A) in yeast lacking a compensatory Hsp70 homolog also causes a growth defect and cell cycle dysregulation after alkylation damage. Together, these data suggest that the dynamic phosphorylation of Hsp70 at this conserved site acts as a regulatory switch to tune chaperone activity in coordination with DNA repair and cell cycle progression.

## Results

### Phosphomimetic Hsc70(T495E) is locked in an open-like conformation

Previous work showed that the *L.p.* kinase LegK4 phosphorylates Hsc70 at a conserved threonine (T495) in its SBD (*Figure 1a and b*; *Moss et al., 2019*). Investigations into the ATPase activity and refolding capacity of in vitro phosphorylated Hsc70 indicated that phosphorylation at T495 reduces its chaperone activity (*Cho et al., 2012*). However, in vitro phosphorylation by LegK4 only yields phosphorylation of ~53% of the Hsc70, complicating bulk biochemical analyses (*Moss et al., 2019*). To overcome this limitation, we generated a phosphomimetic mutant Hsc70(T495E). To determine whether this mutant mimics phosphorylated Hsc70, we assessed the ATPase activity of the purified protein using a malachite green (MG) assay. As Hsc70 has a low intrinsic rate of ATPase activity, we titrated in the co-chaperone DnaJA2 to stimulate hydrolysis (*Moss et al., 2019*; *Russell et al., 1999*). T495E exhibited significantly reduced J-protein-stimulated ATPase activity compared to wild-type Hsc70 (WT) (*Figure 1c*), consistent with prior observations of the LegK4-phosphorylated protein (*Moss et al., 2019*). To investigate if the reduced ATPase activity is due to impaired nucleotide binding, we performed a fluorescence polarization (FP) assay using fluorescently labeled ATP-FAM.

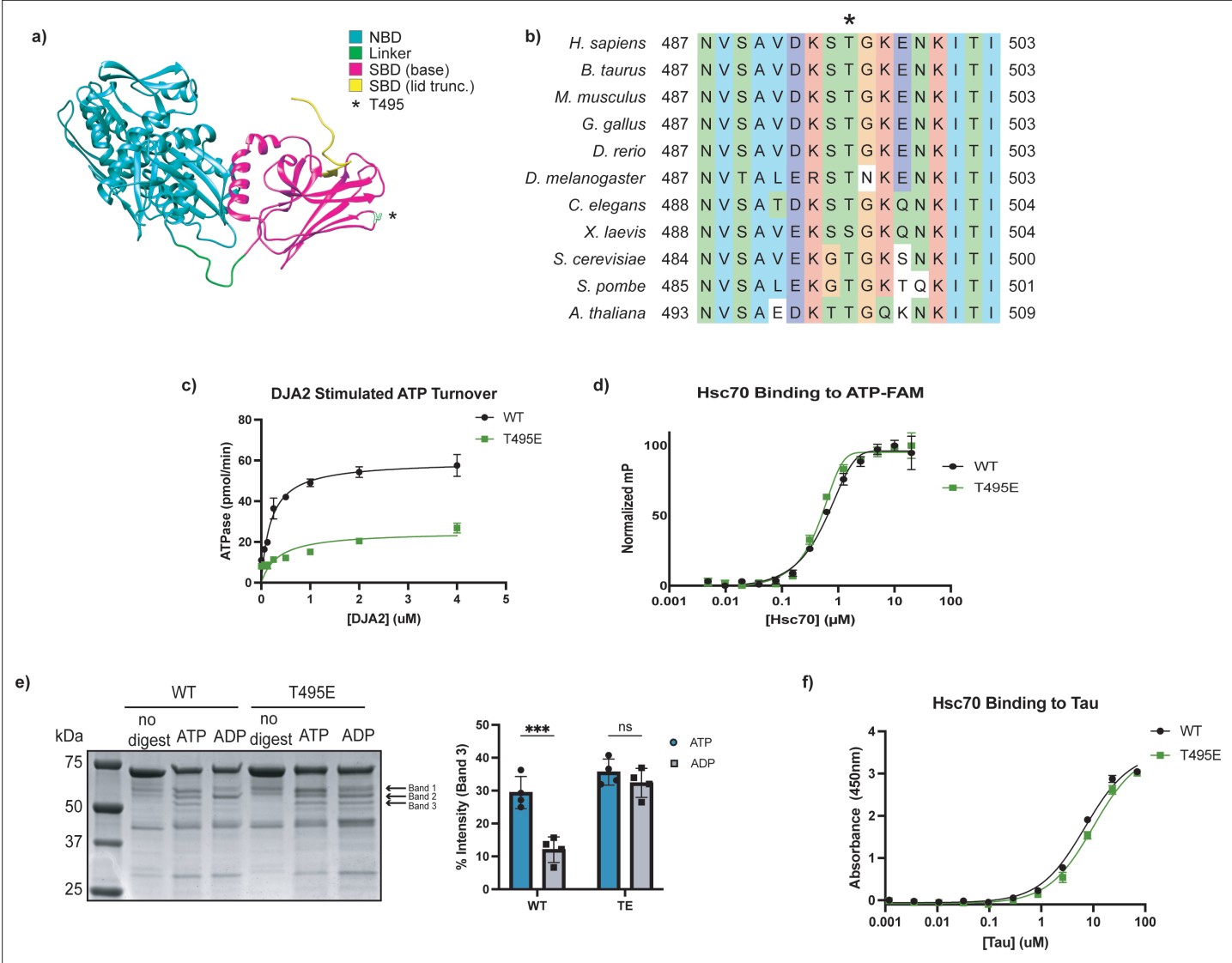

**Figure 1.** The phosphomimetic Hsc70 T495E mutant adopts an open-like conformation. (**a**) Crystal structure of bovine Hsc70 (residues 1–554; PDB: 1YUW). The nucleotide-binding domain (NBD, residues 1–383) is shown in cyan, the interdomain linker (residues 384–394) in green, and the substrate-binding domain (SBD, residues 395–506) in magenta. T495 is highlighted in white and marked with an asterisk. (**b**) Multiple sequence alignment of cytosolic Hsp70 orthologs. Human (P11142), mouse (P63017), cow (P19120), chicken (O73885), zebrafish (Q90473), fruit fly (P11147), worm (P09446), budding yeast (P10591), fission yeast (Q10265), and *Arabidopsis* (P22953) Hsc70 UniProt sequences were aligned with MAFFT (**v7**). Coloring follows CLUSTAL conventions; T495 is marked with an asterisk (*). (**c**) J-protein-stimulated ATPase activity of wild-type (WT) and phosphomimetic Hsc70(T495E) measured by malachite green assay. Data points represent mean + SD of technical triplicates. (**d**) Fluorescence polarization of ATP-FAM binding to WT and T495E Hsc70. Values were normalized to the minimum and maximum polarization; data points represent mean + SD of technical triplicates. (**e**) Partial proteolysis of WT and T495E Hsc70 by trypsin in the presence of ATP or ADP. Digestion products were resolved by SDS-PAGE and visualized with a Coomassie stain, and band intensities were quantified (bar graph, right). Statistical significance was determined by two-way ANOVA with Šidák's multiple comparisons test (adjusted p=0.0002, n=4 technical replicates). (**f**) Tau binding to immobilized WT and T495E Hsc70 measured by ELISA. Data points represent mean + SD of technical triplicates.

The online version of this article includes the following source data for figure 1:

**Source data 1.** tif file containing original unedited Coomassie for *Figure 1e*.

**Source data 2.** tif file containing original Coomassie for *Figure 1e* with bands labeled.

ATP binding was not diminished in T495E (*Figure 1d*), suggesting that T495E allosterically inhibits J protein-stimulated ATPase activity without preventing nucleotide binding.

Hsp70 chaperone activity is driven by cycling through nucleotide-dependent conformations (*Rosenzweig et al., 2019*). In the ATP-bound state, Hsp70s adopt an 'open' conformation, and ATP

hydrolysis to ADP induces large-scale structural rearrangements to a 'closed' state (*Rosenzweig et al., 2019*). As T495E poorly hydrolyzes ATP, we reasoned that this mutation may lock the protein into either the open or closed conformation. Indeed, AMPylation of BiP at the analogous residue is known to lock the protein in an open state (*Preissler et al., 2017a*). To assess the conformation of T495E Hsc70, we performed partial proteolysis using trypsin, which produces distinct cleavage patterns depending on the protein's conformation (*Rinaldi et al., 2018*; *Buchberger et al., 1995*). As expected, WT Hsc70 displayed nucleotide-dependent banding: protection of band 2 and loss of band 3 in the presence of ADP compared to ATP. In contrast, T495E exhibited an ATP-like banding pattern regardless of nucleotide, consistent with a locked open conformation (*Figure 1e*). While this agrees with the studies on BiP AMPylation, our previous work showed that phosphorylation coincides with increased occupancy of Hsc70 on polysomes, suggesting continued substrate engagement (*Moss et al., 2019*). We therefore asked whether T495E can still bind client proteins. An ELISA with tau, a known Hsc70 substrate (*Kundel et al., 2018*), showed that T495E retains the ability to bind tau despite the conformational lock (*Figure 1f*). These results suggest that T495E locks Hsc70 in a pseudo-open conformation: structurally similar to the ATP-bound open state in terms of protease sensitivity and domain exposure, yet still capable of substrate engagement typically associated with the closed state.

## BER leads to Hsp70(T495) phosphorylation

Pathogens often mimic host proteins and hijack cellular pathways to create a favorable niche for replication. Thus, these pathogen-driven modifications frequently reveal biologically significant regulatory nodes within host systems (*Lee and Machner, 2018*). Given this tendency of pathogens and the high degree of conservation of T495 (*Figure 1b*), we reasoned that *L.p.* phosphorylation of Hsp70 may be targeting an important regulatory module. Motivated by this concept, we examined published phospho-proteomics datasets to determine if this phosphorylation had been previously observed in eukaryotes. A phospho-proteomics study in *S. cerevisiae* identified phosphorylation at the analogous residue, Ssa1(T492), after treatment with the alkylating agent methyl methanesulfonate (MMS) (*Albuquerque et al., 2008*). Inspired by this finding, we analyzed human cells treated with MMS and observed that this phosphorylation is conserved (*Figure 2a*). As expected, this band is sensitive to phosphatase activity (*Figure 2—figure supplement 1a*).

This finding suggested that phosphorylation of Hsp70 might be involved in the response to DNA damage. MMS is primarily used as a DNA alkylating agent, though its chemical activity is not limited to DNA targets (*Thomas et al., 2020*). Thus, we sought to determine whether the response to MMS-induced DNA alkylation leads to Hsp70 phosphorylation. Aberrant DNA alkylation is predominantly repaired through the BER pathway (*Beard et al., 2019*). In BER, DNA glycosylases recognize and excise damaged bases, producing an abasic site (AP site). MMS-induced lesions are processed by N-methylpurine DNA glycosylase (MPG) (*Beard et al., 2019*; *Sedgwick, 2004*; *Figure 2b*). The backbone at the AP site is then cleaved by the endonuclease APE1, and the resultant single-stranded break is repaired by a variety of proteins, including the DNA polymerase Polβ (*Beard et al., 2019*; *Figure 2b*). Interestingly, the predominant DNA adduct generated by MMS, 7-methylguanine, is not intrinsically cytotoxic, but spontaneously depurinates to a toxic and mutagenic intermediate and so requires rapid repair (*Wyatt and Pittman, 2006*; *Fu et al., 2012*). The repair intermediates generated by the BER pathway, however, are cytotoxic (*Wyatt and Pittman, 2006*; *Sobol et al., 2003*). Because Polβ activity is rate-limiting, overexpression of the glycosylase that initiates BER can lead to an accumulation of these cytotoxic intermediates (*Beard et al., 2019*; *Srivastava et al., 1998*). To determine whether increased BER activity affects Hsp70 phosphorylation, we treated cells overexpressing MPG with MMS. MPG overexpression modestly increased MMS-induced pHsp70 levels (*Figure 2c*). Conversely, inhibition of the subsequent BER step strongly reduced the levels of MMS-induced pHsp70. Pharmacological inhibition of APE1 led to a dose-dependent decrease in Hsp70 phosphorylation (*Figure 2d*). Similarly, treatment with methoxyamine (Mx), which covalently binds AP sites and impairs APE1 cleavage and Polβ activity (*Horton et al., 2000*), caused a striking reduction in pHsp70 levels (*Figure 2b and e*). Surprisingly, overexpression of APE1 did not enhance pHsp70 (*Figure 2—figure supplement 1b*), suggesting that APE1 activity is necessary but not rate-limiting in this context. Additionally, overexpression of Polβ did not reduce pHsp70 upon MMS treatment (*Figure 2—figure supplement 1c*). These data confirm that the DNA alkylation by MMS is responsible

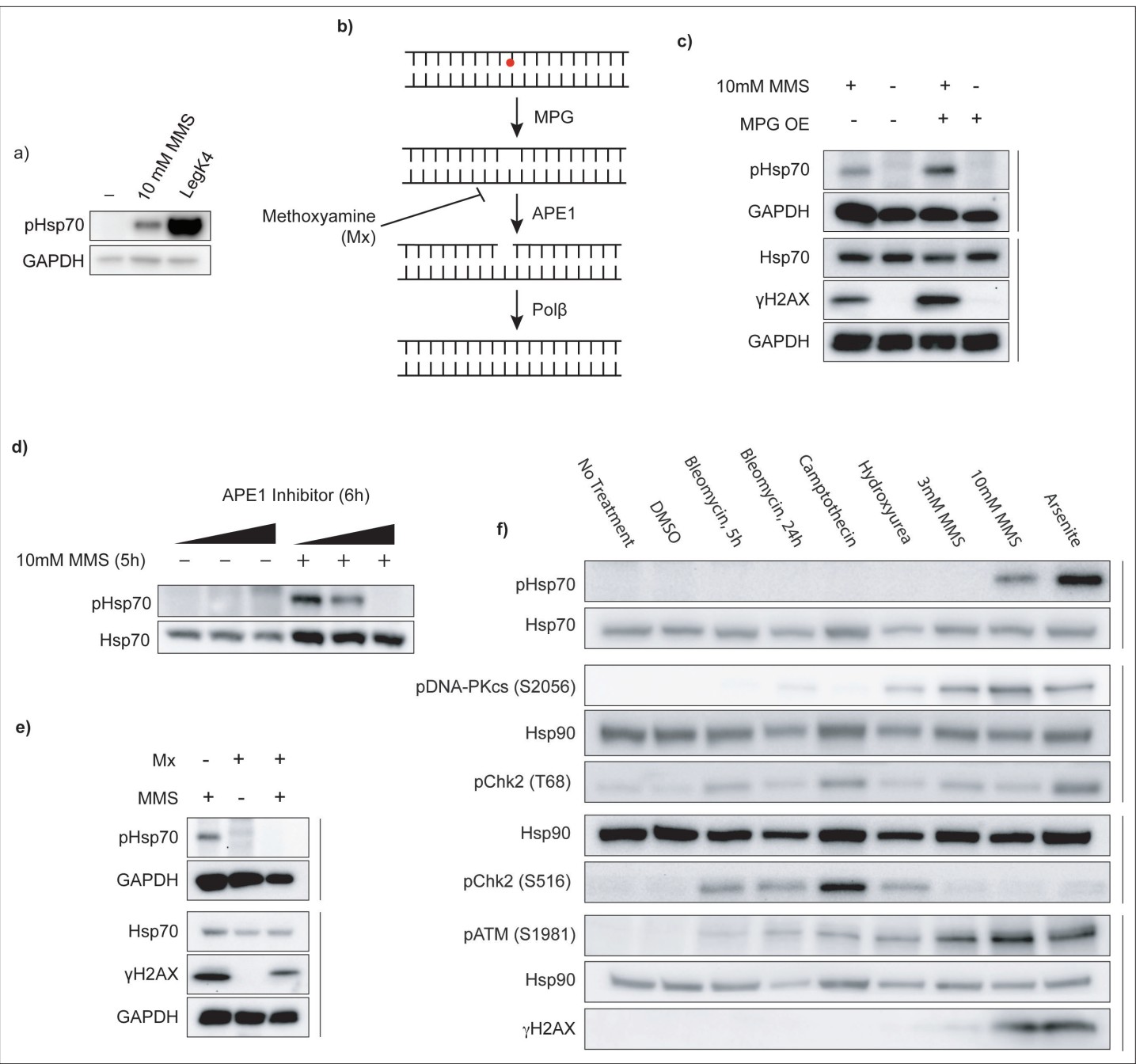

**Figure 2.** Base excision repair drives phosphorylation of Hsp70 in human cells. (**a**) Hsp70 phosphorylation by methyl methanesulfonate (MMS) treatment or LegK4 overexpression. HEK293T cells were transiently transfected overnight with LegK4Δ1–58:GFP or treated with 10 mM MMS for 5 hr. Phosphorylation at T495 was detected using a phospho-specific antibody to pHsp70 T495; GAPDH was used as a loading control. Data are representative of n>3 independent experiments. (**b**) Schematic of the base excision repair (BER) pathway, highlighting steps relevant to MMS-induced DNA damage. (**c**) N-methylpurine DNA glycosylase (MPG) overexpression increases pHsp70 levels. HEK293T cells were transiently transfected with MPG overnight, treated with MMS, and analyzed by immunoblotting for pHsp70, the loading controls GAPDH and Hsp70, and the DNA damage marker γH2AX. Data are representative of n=3 independent experiments. (**d**) Inhibition of APE1 reduces MMS-induced pHsc70. Cells were pretreated for 1 hr with 5 μM, 10 μM, or 50 μM APE1 inhibitor (APE1 compound III) before MMS treatment. Hsp70 and pHsp70 were detected by immunoblotting. Data are representative of n=3 independent experiments. (**e**) Masking of AP sites prevents pHsp70 accumulation. Cells were pretreated with 60 mM methoxyamine (Mx) for 30 min, followed by cotreatment with 30 mM Mx and 10 mM MMS for 5 hr. pHsp70, the loading controls GAPDH and Hsp70, and the DNA damage marker γH2AX were detected by immunoblotting. Data are representative of n=3 independent experiments. (**f**) DNA damage specificity panel for pHsp70 induction. Cells were treated with bleomycin (10 μM, 5 hr or 24 hr), camptothecin (10 μM, 5 hr), hydroxyurea (2 mM, 24 hr), MMS (3 mM or 10 mM, 5 hr), sodium arsenite (0.5 mM, 5 hr), or DMSO vehicle (5 hr). Immunoblotting was performed for pHsp70, DDR kinase activation

*Figure 2 continued on next page*

*Figure 2 continued*

markers (pDNA-PKcs S2056, pChk2 T68, pChk2 S16, pATM S1981), DNA damage marker γH2AX, and loading controls (Hsp70 and Hsp90). Data are representative of n=3 independent experiments.

The online version of this article includes the following source data and figure supplement(s) for figure 2:

**Source data 1.** tif files containing original unedited Western blots for *Figure 2a and c–f*.

**Source data 2.** tif file containing original Western blots for *Figure 2a and c–f* with bands labeled.

**Figure supplement 1.** Base excision repair drives phosphorylation of Hsp70 in human cells.

**Figure supplement 1—source data 1.** tif files containing original unedited Western blots for *Figure 2—figure supplement 1a–e*.

**Figure supplement 1—source data 2.** tif file containing original Western blots for *Figure 2—figure supplement 1a–e* with bands labeled.

for Hsp70 phosphorylation in human cells, and the results specifically indicate that BER intermediates may trigger this response.

To test the specificity of Hsp70 phosphorylation in response to DNA damage, we treated cells with a panel of genotoxic compounds that activate distinct branches of the DDR, including bleomycin (induces double-stranded breaks [DSBs]), camptothecin (a topoisomerase I inhibitor), hydroxyurea (depletes the dNTP pool), MMS, and arsenite (generates reactive oxygen species). While activation of DDR kinases (pDNA-PKcs(S2056), pATM(S1981), pChk2(T68), pChk2(S516)), and an increase in the DDR marker γH2AX confirm DNA damage and response induction in the drug treatment conditions, pHsp70 was only observed following high-dose MMS treatment and exposure to sodium arsenite (*Figure 2f*). Arsenite causes oxidative DNA damage that can be repaired through BER (*Beard et al., 2019*; *Bau et al., 2002*). Surprisingly, arsenite has also been shown to both inhibit the activity of enzymes necessary for this pathway, including OGG (the oxidative glycosylase) (*Ebert et al., 2011*), and decrease the levels of others (e.g. APE1 and Polβ) (*Sykora and Snow, 2008*). Despite this, we found that both treatment with Mx and inhibition of APE1 were still sufficient to prevent Hsp70 phosphorylation upon arsenite treatment (*Figure 2—figure supplement 1d and e*). These findings suggest that Hsp70(T495) phosphorylation is selectively triggered by repair intermediates generated during BER.

## Phosphorylation of Hsp70 requires the DDR kinases ATM, DNA-PKcs, Chk2, and CK1

APE1 activity generates cytotoxic adducts. Its endonuclease activity at abasic sites directly generates single-stranded DNA breaks (SSBs) (*Caldecott, 2008*). These SSBs can, in turn, convert to DSBs through two primary mechanisms: collision of a replication fork with an SSB (*Caldecott, 2008*; *Nikolova et al., 2010*; *Ensminger et al., 2014*) or generation of SSBs in proximity on opposing strands of DNA (*Polyzos et al., 2024*). In these cases, damaged DNA is sensed and a response subsequently mounted through DDR pathways. At the highest level, the varied branches of the DDR are controlled by three master kinases: ataxia-telangiectasia mutated (ATM), ataxia-telangiectasia and Rad3-related (ATR), and DNA-dependent protein kinase (DNA-PK). ATR orchestrates the response to SSBs, while ATM and DNA-PK direct DSB repair through homologous recombination and non-homologous end joining, respectively (*Giglia-Mari et al., 2011*; *Durocher and Jackson, 2001*). While this broadly describes the organization of the DDR, the apparent simplicity belies the fact that these master regulators induce varied and nuanced responses depending on the exact type and severity of DNA damage present (*Lowndes and Murguia, 2000*; *Caron et al., 2015*; *Heijink et al., 2013*; *Lanz et al., 2019*).

To determine if DDR signaling contributes to MMS-induced Hsp70 phosphorylation, we performed siRNA-mediated knockdowns of these three master kinases. Among these, only knockdown of DNA-PKcs prevented Hsp70 phosphorylation (*Figure 3a*, *Figure 3—figure supplement 1a, b*). Interestingly, while ATM knockdown had no effect, pharmacological inhibition of ATM decreased Hsp70 phosphorylation (*Figure 3b*). To rule out off-target effects, we tested two independent ATM inhibitors and observed a similar reduction in pHsp70 with both (*Figure 3b*). This discrepancy is consistent with prior reports showing that genetic and pharmacological inhibition of ATM can yield diverging results (*Choi et al., 2010*; *Menolfi and Zha, 2020*). Pharmacological inhibition of DNA-PKcs confirmed the DNA-PKcs knockdown experiment (*Figure 3c*). Both DNA-PKcs and ATM phosphorylate the transducer kinase Chk2, activating it and promoting propagation of the DDR (*Zannini et al., 2014*). Inhibition of Chk2 also prevented Hsp70 phosphorylation

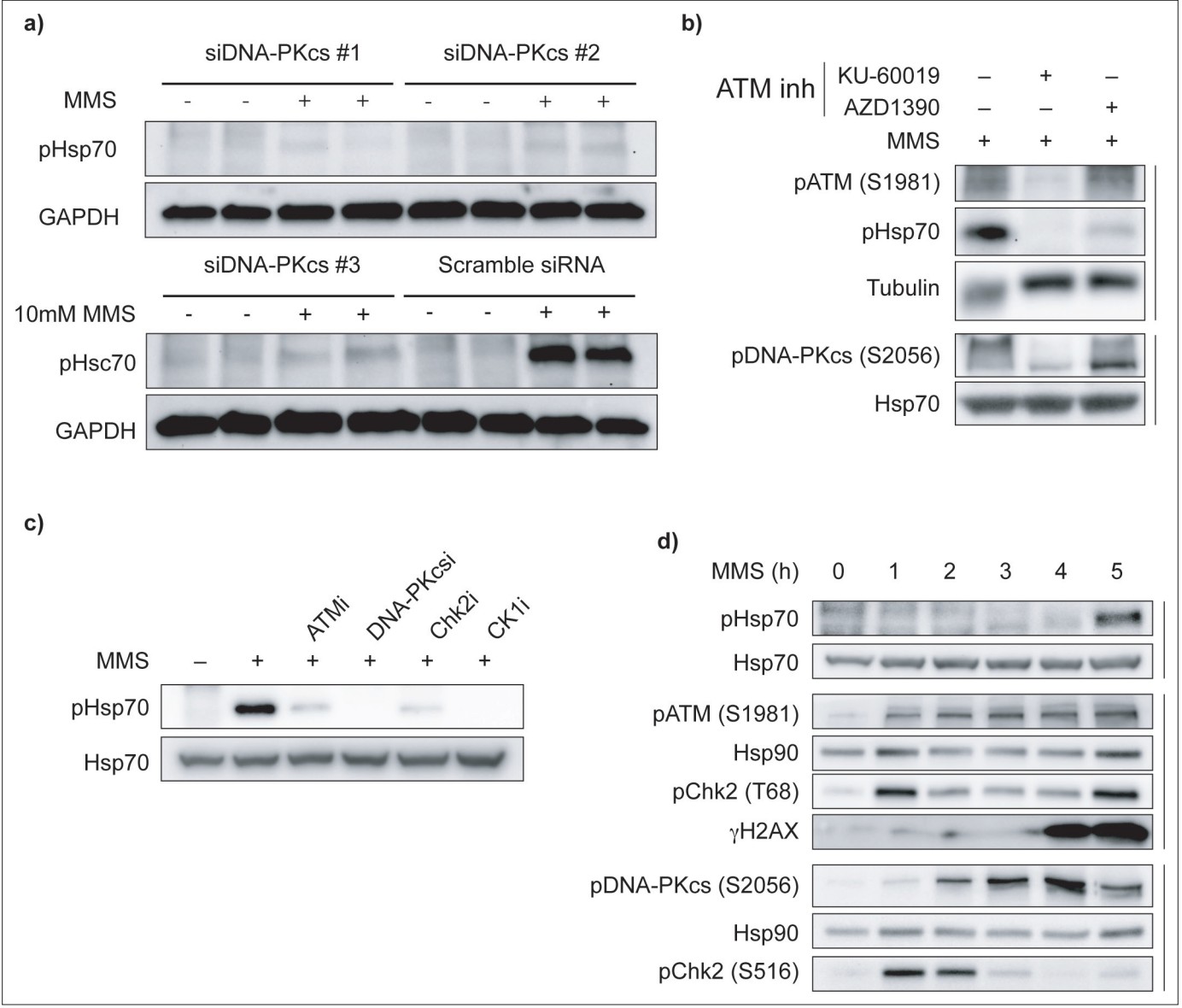

**Figure 3.** DNA damage response (DDR) kinase activity is upstream of Hsp70 phosphorylation. (**a**) DNA-PKcs knockdown reduces pHsp70 levels. Cells were transiently transfected with three independent siRNAs targeting DNA-PKcs or a scramble control for 72 hr, followed by treatment with 10 mM methyl methanesulfonate (MMS) for 5 hr. pHsp70 and GAPDH (loading control) were detected by immunoblotting. Data are representative of n=3 independent experiments. (**b**) Pharmacological inhibition of ataxia-telangiectasia mutated (ATM) decreases pHsp70 induction. Cells were pretreated for 1 hr with ATM inhibitors (10 μM KU-60019 or 200 nM AZD1390), then treated with 10 mM MMS for 5 hr. ATM and DNA-PKcs autoactivation were monitored by immunoblotting pATM (S1981) and pDNA-PKcs (S2056), respectively. Tubulin and total Hsp70 served as loading controls. Data are representative of n=3 independent experiments. (**c**) Pharmacological inhibition of ATM, DNA-PKcs, Chk2, and CK1 decreases Hsp70 phosphorylation during MMS treatment. Cells were pretreated for 1 hr with inhibitors for ATM (200 nM AZD1390), DNA-PKcs (2 μM AZD7648), Chk2 (5 μM CCT241533), CK1 (50 μM PF-670462), or with vehicle control (DMSO), then treated with 10 mM MMS for 5 hr. Immunoblotting was performed against pHsp70 and the loading control Hsp70. Data are representative of n=3 independent experiments. (**d**) Time course of MMS-induced DDR activation and pHsp70 phosphorylation. Cells were treated with 10 mM MMS and harvested hourly. ATM and DNA-PKcs activation were detected by pATM (S1981) and pDNA-PKcs (S2056), respectively. Chk2 activation was monitored by pChk2 (**T68**) and pChk2 (S516). DNA damage was assessed via γH2AX. Hsp70 and Hsp90 were used as loading controls. Data are representative of n=3 independent experiments.

The online version of this article includes the following source data and figure supplement(s) for figure 3:

**Source data 1.** tif files containing original unedited Western blots for *Figure 3a–d*.

**Source data 2.** tif file containing original Western blots for *Figure 3a–d* with bands labeled.

**Figure supplement 1.** DNA damage response (DDR) kinase activity is upstream of Hsp70 phosphorylation.

*Figure 3 continued on next page*

*Figure 3 continued*

**Figure supplement 1—source data 1.** tif files containing original unedited Western blots for *Figure 3—figure supplement 1a–d*.

**Figure supplement 1—source data 2.** tif file containing original Western blots for *Figure 3—figure supplement 1a–d* with bands labeled.

(*Figure 3c*). Given that casein kinase 1 (CK1) has been previously reported to act in concert with Chk2 in the DDR (*Tuppi et al., 2018*), we tested its involvement and found that CK1 inhibition also suppressed Hsp70 phosphorylation (*Figure 3d*, *Figure 3—figure supplement 1c*). Inhibition of these kinases also prevented Hsp70 phosphorylation upon arsenite treatment (*Figure 3—figure supplement 1d*).

While these kinases appear to be necessary for Hsp70 phosphorylation, our data are more consistent with their involvement in upstream signaling rather than directly acting on Hsp70. We would further highlight that the delay in Hsp70 phosphorylation contrasts with canonical DDR signaling, which is initiated within minutes of damage induction (*Canman et al., 1998*; *Chou et al., 2015*), pHsp70 only emerges after prolonged MMS exposure (*Figure 3d*). Indeed, we observed robust activation of DNA-PKcs, ATM, and Chk2 well before Hsp70 phosphorylation (*Figure 3d*). These observations raise the possibility that either prolonged MMS exposure or a secondary cellular response is required to trigger Hsp70 phosphorylation.

## Hsp70 phosphorylation occurs after M phase onset

To better understand the timing of Hsp70 phosphorylation, we treated cells with MMS for varying durations (1–5 hr), followed by recovery in fresh media up to a total of 5 hr. In previous experiments, we observed robust Hsp70 phosphorylation only after 5 hr of continuous MMS treatment, with minimal signal at earlier time points (see *Figure 3d*). However, in our pulse-chase assay, 2 hr MMS treatment, when followed by a 3 hr MMS-free chase, resulted in pHsp70 accumulation (*Figure 4a*). This result indicates that prolonged MMS exposure alone does not fully explain the lag time between damage initiation and the appearance of pHsp70, suggesting that secondary cellular events or signaling contribute to Hsp70 phosphorylation.

Damage from MMS is thought to occur when replication forks collide with DNA repair intermediates in S phase (*Nikolova et al., 2010*; *Ensminger et al., 2014*). This hypothesis, in combination with our observations that pHsp70 accumulation lags behind initial DDR activation, suggested that cell cycle stage might be a key determinant in Hsp70 phosphorylation. In asynchronous populations, Hsp70 phosphorylation might only appear when enough cells reach a permissive phase of the cell cycle. If this were true, synchronizing cells at the beginning of S phase should potentiate MMS-induced Hsp70 phosphorylation. To test this idea, we synchronized cells at the beginning of S phase using a double thymidine block, then released them into MMS-containing or untreated media. Unexpectedly, synchronization at G1/S does not increase Hsp70 phosphorylation (*Figure 4b*). Even more surprising, MMS-treated cells entered mitosis at an increased rate compared to controls, as evidenced by the accumulation of the mitotic marker phospho-histone H3 (S10) (pH3), reduced levels of S phase markers such as the replication licensing factor CDT1 and thymidine kinase, and pCdk1(Y15), the inhibited cyclin-dependent kinase that must be de-phosphorylated for M phase entry (*Figure 4b*). These data suggest that MMS drives cells into mitosis, and that Hsp70 phosphorylation occurs after this transition. Indeed, blocking mitotic entry by treatment with a CDK1 inhibitor (Ro3306) prevented both MMS and arsenite-induced Hsp70 phosphorylation (*Figure 4c*, *Figure 4—figure supplement 1a*). Releasing the CDK1 block upon MMS or arsenite treatment restored Hsp70 phosphorylation (*Figure 4c*, *Figure 4—figure supplement 1a*), suggesting this response does not require passage through S phase. While passage through mitosis upon DNA damage is necessary for Hsp70 phosphorylation, the presence of pHsp70 in a distinct nuclear fraction indicates that it persists after nuclear envelope reformation (*Figure 4d*). Indeed, a 2 hr pulse chase of MMS revealed that pHsp70 accumulates even after pH3 disappears, indicating that phosphorylation is maintained after mitotic exit (*Figure 4e*). Notably, we do not see an increase in S phase markers such as thymidine kinase, nor changes in cyclin A or cyclin B levels, suggesting that the cell cycle progression may be dysregulated following MMS treatment.

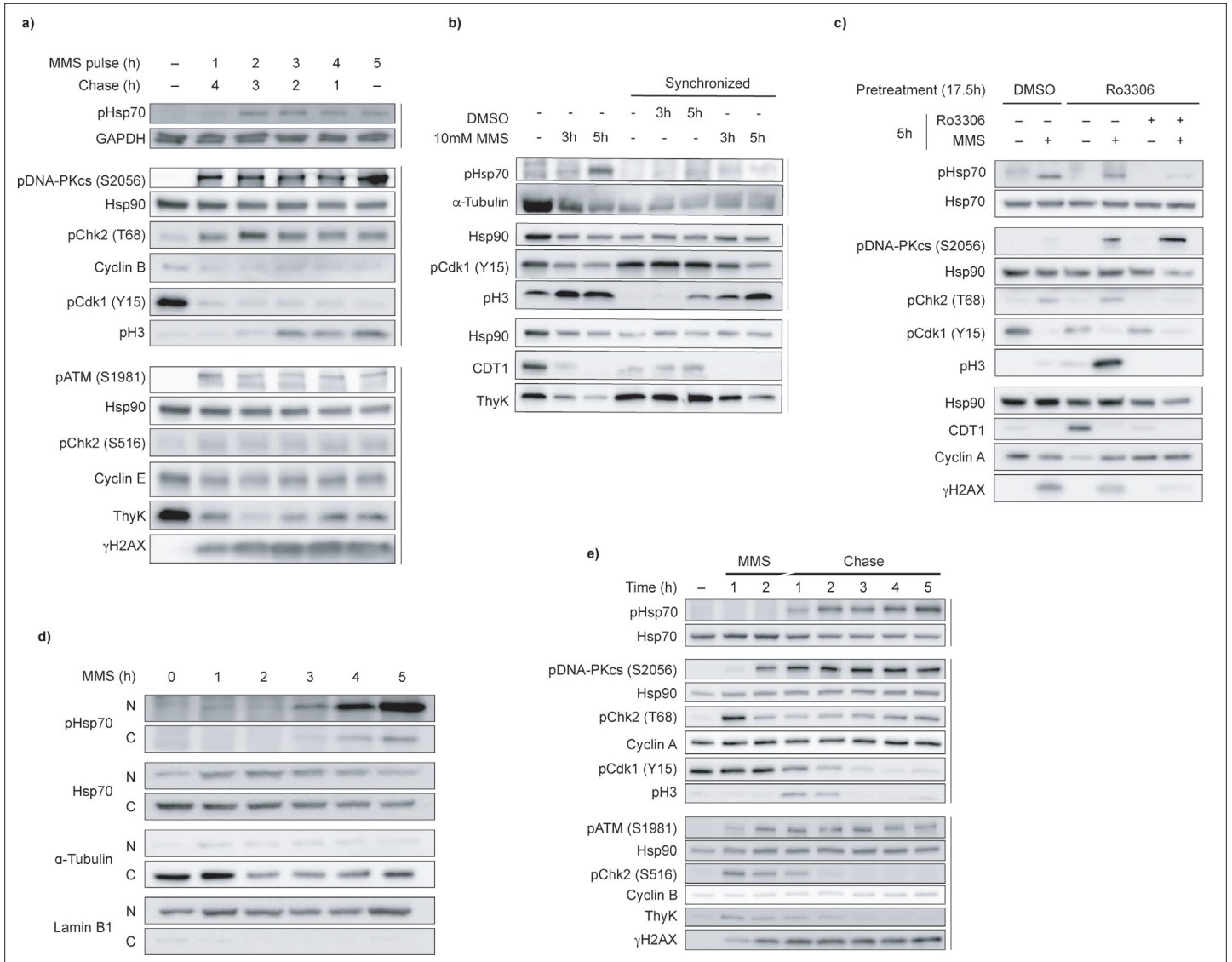

**Figure 4.** Hsp70 phosphorylation is linked to the cell cycle. (**a**) Variable pulse-chase methyl methanesulfonate (MMS) treatment suggests a complex signaling pathway. Cells were treated with 10 mM MMS for 1–5 hr, washed, then incubated in MMS-free media for the remainder of the 5 hr period. Ataxia-telangiectasia mutated (ATM) and DNA-PKcs activation were detected by pATM (S1981) and pDNA-PKcs (S2056); Chk2 activation by pChk2 (T68) and pChk2 (S516); DNA damage by γH2AX. Cell cycle progression was monitored with S phase markers thymidine kinase (ThyK) and CDT1; mitotic entry marker pCdk1(Y15) (dephosphorylation permits M phase entry), M phase marker phospho-histone H3 (**S10**) (pH3); and cyclins B and E. GAPDH and Hsp90 are loading controls. Data are representative of n=3 independent experiments. (**b**) Early S phase synchronization fails to increase pHsp70 accumulation. Cells were synchronized with 2.5 mM thymidine (18 hr→ 9 hr release → 17 hr retreatment) and released into fresh media ±10 mM MMS for the indicated times. Asynchronous cells were also MMS-treated. Immunoblots were performed for pHsp70, cell cycle markers (pCdk1 Y15, pH3, CDT1, ThyK), and loading controls α-tubulin and Hsp90. Data are representative of n=2 independent experiments. (**c**) G2/M stalling by CDK1 inhibition reduces pHsp70 levels. Cells were pretreated with 10 µM CDK1 inhibitor Ro3306 or DMSO for 17.5 hr, washed, then treated again with Ro3306 or DMSO ± 10 mM MMS for 5 hr. Immunoblotting was performed for pHsp70, cell cycle markers (pCdk1 Y15, pH3, CDT1, cyclin A), DDR markers (pDNA-PKcs S2056, pChk2 T68, γH2AX), and loading controls Hsp70 and Hsp90. Data are representative of n=3 independent experiments. (**d**) Subcellular fractionation of pHsp70 during MMS treatment shows nuclear localization. Cells were treated with 10 mM MMS from 1 to 5 hr or left untreated. Cytoplasmic and nuclear extracts were prepared using the NE-PER kit. Immunoblotting was performed for pHsp70 and total Hsp70 levels; α-tubulin and lamin B1 served as cytoplasmic and nuclear markers, respectively. Data are representative of n=3 independent experiments. (**e**) A 2 hr MMS pulse chase reveals pHsp70 accumulation post-mitosis. Cells were treated with 10 mM MMS for 2 hr, washed, and then incubated in fresh media. Samples were harvested hourly, alongside an untreated control. Immunoblotting was performed for pHsp70, DDR markers (pDNA-PKcs S2056, pChk2 T68, pATM S1981, pChk2 S516, γH2AX), cell cycle markers (cyclin A, pCdk1 Y15, pH3, cyclin B, ThyK), and loading controls Hsp70 and Hsp90. Data are representative of n=3 independent experiments.

The online version of this article includes the following source data and figure supplement(s) for figure 4:

*Figure 4 continued on next page*

*Figure 4 continued*

**Source data 1.** tif files containing original unedited Western blots for *Figure 4a–e*.

**Source data 2.** tif file containing original Western blots for *Figure 4a–e* with bands labeled.

**Figure supplement 1.** Mitosis precedes Hsp70 phosphorylation.

**Figure supplement 1—source data 1.** tif files containing original unedited Western blots for *Figure 4—figure supplement 1a*.

**Figure supplement 1—source data 2.** tif file containing original Western blots for *Figure 4—figure supplement 1a* with bands labeled.

## Phosphoregulation of Ssa1 at T492 is important for G1/S transition in yeast

To investigate the functional significance of Hsp70 phosphorylation at T495, we turned to *S. cerevisiae* due to its genetic tractability. *S. cerevisiae* has four cytosolic Hsp70s, Ssa1-4. Of these, Ssa1 and Ssa2 are constitutively expressed, whereas Ssa3 and Ssa4 are stress-inducible (*Werner-Washburne et al., 1987*). Phosphoproteomics studies revealed that Ssa1 can be phosphorylated at threonine 492, which corresponds to human Hsp70 T495 (*Albuquerque et al., 2008*; *Holt et al., 2009*). To probe the role of this modification, we created phosphomimetic (T492E, TE) and phosphonull (T492A, TA) point mutations at the endogenous *SSA1* locus. Given the known redundancy between Ssa1 and Ssa2 (*Craig and Jacobsen, 1984*), we disrupted *SSA2* by inserting an antimicrobial resistance gene into its coding sequence to better assess the functional consequences of Ssa1 phospho-variants. To confirm the conservation of Hsp70 phosphorylation in yeast, we treated our mutants with MMS as previously described (*Albuquerque et al., 2008*) and performed a Western blot. Notably, we see faint pHsp70 signal in WT yeast in the absence of MMS treatment (*Figure 5a*). This corresponds with prior work indicating low levels of phospho-Ssa1 (T492) in asynchronously growing cells (*Holt et al., 2009*). This signal is diminished in the phosphomutant strains, and in all *ssa2Δ* strains, both supporting the conclusion that our antibody binds pHsp70(T492) in yeast, and indicating that there is likely redundant phospho-regulation of Ssa2 as well. Additionally, we find that MMS treatment leads to a strong increase in Hsp70 phosphorylation in WT, and phosphorylation to a lesser extent in WT;*ssa2Δ* (*Figure 5a*). Strikingly, even in the *SSA2* background, we observed a growth defect in TE yeast (*Figure 5b and c*). This is not likely explained by hypomorphism, as neither the TA mutant nor the WT;*ssa2Δ* strains show such a defect. Rather, phosphomimetic Ssa1 seems to act in a dominant-like manner. In the *ssa2Δ* background, strains expressing either phosphomimetic or phosphonull Ssa1 grew more slowly than the wild-type (*Figure 5b and c*).

Phosphorylation of Ssa1 at T492 was previously reported to occur in a cell cycle-dependent manner, specifically when yeast are prevented from exiting mitosis through the expression of a non-degradable cyclin B (*Holt et al., 2009*). This information, in conjunction with the cell cycle dependence of pHsp70 in human cells and the growth defects we observed in our yeast phosphomutants, led us to examine the impact of these mutations on cell cycle progression. With SSA2 present, the Ssa1 WT and phosphonull (TA) strains showed similar cell cycle distributions. In contrast, the phosphomimetic mutant (TE) showed an increase in cells with 1N DNA content, consistent with a G1 stall (*Figure 5d*). In the *ssa2Δ* background for both Ssa1 WT and TA, we see a slight increase in the G1 population compared to G2, and the difference is even more pronounced in T492E yeast (*Figure 5d*). T492A yeast showed increased cell size in G1 in the *ssa2Δ* background, a phenotype previously linked to G1/S stalling, perhaps due to decreased Cln3 stability or disrupted Cln3-Ssa1 (*Truman et al., 2012*; *Moreno et al., 2019*; *Figure 5—figure supplement 1a*). Intriguingly, both *ssa2Δ* point mutants exhibited a large cell size in G2/M (*Figure 5—figure supplement 1a*) – a finding that warrants additional investigation. Together, these data suggest that dynamic phospho-regulation of Ssa1 at T492 is critical for proper cell cycle progression.

We were curious if this modification served a similar cell cycle regulatory function during MMS treatment. To assess the overall viability of the point mutants in response to alkylation damage, we performed a spot test in the presence and absence of MMS. In the *ssa2Δ* background, both phosphonull and phosphomimetic Ssa1 led to impaired growth on MMS-containing media, though growth in the *SSA2* background was only mildly impacted (*Figure 5e*). To examine how these mutations influence cell cycle progression in response to MMS, we treated the strains with 0.05% MMS for 3 hr, as previously shown to cause Ssa1 phosphorylation at T492 (*Figure 5a*; *Albuquerque et al., 2008*), and released them into fresh YPAD for the indicated times (*Figure 5f*). Ssa1 WT strains with and without

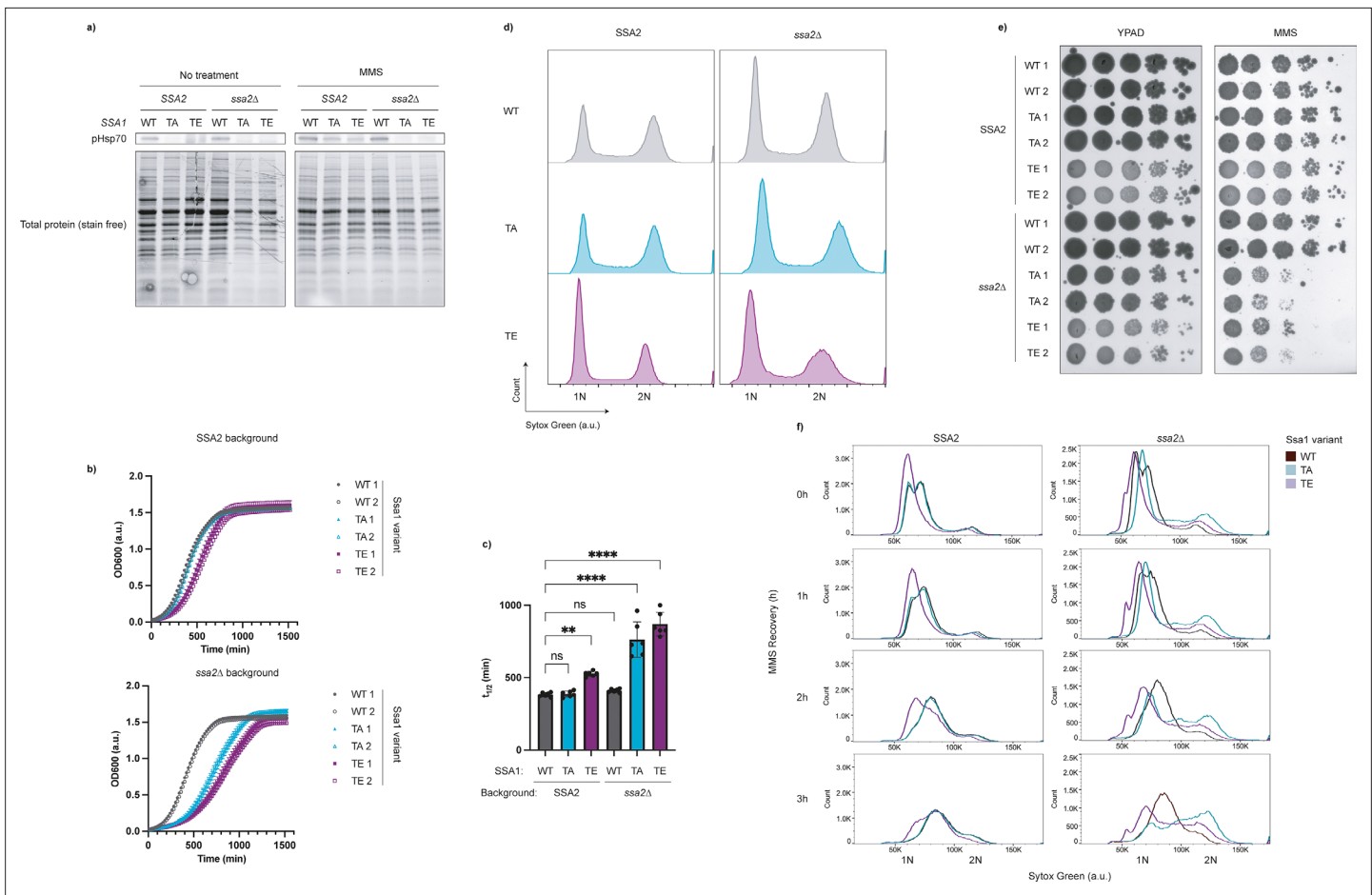

**Figure 5.** Ssa1 T492 phosphorylation mutations cause cell cycle defects in *S. cerevisiae*. (**a**) Phosphorylation of Hsp70 is conserved in yeast. Strains were grown to mid-log and treated with 0.05% methyl methanesulfonate (MMS) in YPAD or fresh YPAD for 3 hr. Immunoblots were performed for pHsp70, and total protein (Stain-Free, Bio-Rad) was used as a loading control. Data are representative of n=2 experiments. (**b**) Growth curves of *S. cerevisiae* Ssa1 mutants show delayed growth. Indicated strains were grown to mid-log phase, diluted to the same starting concentration, and monitored overnight at 30°C in a plate reader. Data represent the average of technical triplicates. Data are representative of n=3 independent experiments. (**c**) Half-times ($t_{1/2}$) of both Ssa1 mutants in the *ssa2Δ* background, and of the phosphomimetic mutant in the *SSA2* background, are significantly increased. Sigmoidal fits were applied to growth curves to determine $t_{1/2}$ values. The data represent three technical replicates with two biological replicates per strain. Bars represent mean ± SD of six replicates (n=6; 2 biological replicates × 3 technical replicates) Statistical significance was determined by ordinary one-way ANOVA followed by Dunnett's multiple comparison test (**p=0.0015; ****p<0.0001). (**d**) Cell cycle distribution analysis reveals G1 stalling of Ssa1 phosphomutants. Yeast were grown to mid-log phase, adjusted to the same concentration, and immediately fixed. Cells were stained with Sytox Green and analyzed by flow cytometry to determine DNA content. Histograms display DNA content (X-axis) with 1N corresponding to G1 phase, 2N to G2/M, and intermediate values to S phase. Left: WT *SSA2* background; right: *ssa2Δ* background. Data are representative of n=4 technical replicates with 2 biological replicates per strain. (**e**) Ssa1 phosphomutants display increased MMS sensitivity in a spot test assay. Yeast were grown to mid-log phase, adjusted to 2e7 cells/mL, serially diluted 1:10, and spotted (5 μL) onto YPAD plates with or without 0.0095% MMS. Plates were incubated at 30°C and imaged after 3 days. Data are representative of n=3 independent experiments. (**f**) Ssa1 phosphomutants exhibit perturbed G1/S stalling during MMS recovery. Yeast were grown to mid-log phase, treated with 0.05% MMS for 3 hr, washed, and resuspended in fresh media for recovery. Samples were collected at the indicated times points and analyzed by staining and flow cytometry as described as in (**d**). Data are representative of n=2 technical replicates with 2 biological replicates per strain.

The online version of this article includes the following source data and figure supplement(s) for figure 5:

**Source data 1.** tif files containing original unedited Western blot and Stain-Free gel for *Figure 5a*.

**Source data 2.** tif file containing original Western blot and Stain-Free gel for *Figure 5a* with bands labeled.

**Figure supplement 1.** Cell size is increased for Ssa1 phosphomutant yeast strains in the absence of Ssa2.

*SSA2* behave similarly: immediately after the MMS treatment, the cells display a bimodal distribution around G1 and early S. Upon MMS release, the G1 peak gradually disappeared, accompanied by synchronized progression through S phase, with DNA content slowly increasing over time. In the *SSA2* background, the T492A was nearly indistinguishable from the WT strains. However, in the *ssa2Δ* background, T492A no longer exhibited the bimodal distribution of cells near 1N immediately after MMS treatment. Rather, there was a single peak. Upon release, the cells failed to undergo coordinated S phase progression. Instead, DNA content increased in an asynchronous manner, suggesting impaired checkpoint regulation. The T492E mutants also showed distinct phenotypes. In the *SSA2* background, T492E cells lacked the bimodal distribution around 1N immediately after MMS treatment and instead exhibited a more pronounced G1 arrest. Upon release, these cells progressed through S phase more slowly than WT, with a prominent left shoulder, indicating delayed DNA replication in a subset of cells. In the *ssa2Δ* background, the phenotype was even more prominent. A strong 1N peak persisted after MMS release, with minimal evidence of bulk S phase progression. Instead, we observed a gradual increase in DNA content across the population, reminiscent of the T492A mutant in *ssa2Δ*, but with a key difference: while T492A;*ssa2Δ* cells showed more rapid entry to and passage through S phase, T492E;*ssa2Δ* cells retained a distinct 1N peak and slower S phase progression. These data indicate that both phosphonull and phosphomimetic mutations disrupt the coordination of cell cycle re-entry following genotoxic stress. Loss of phosphorylation appears to weaken G1 arrest and promote uncontrolled S phase entry, while constitutive phosphorylation reinforces arrest but impairs orderly progression through S phase. Eliminating the functional redundancy of Ssa2 in the T492A;*ssa2Δ* and T492E;*ssa2Δ* strains reveals the importance of dynamic Ssa1 phospho-regulation in mounting an effective DDR and ensuring proper cell cycle transitions**,** as both phosphonull and phosphomimetic Ssa1 variants result in cell cycle dysregulation and altered growth in the absence of Ssa2.

## Discussion

Our work demonstrates that Hsp70s are phosphorylated at T495 as part of a conserved response to DNA damage and that dynamic phosphoregulation of this residue exerts control over the cell cycle. While prior phosphoproteomics studies in yeast detected phosphorylation of the analogous residue in Ssa1 (T492) following MMS exposure (*Albuquerque et al., 2008*) or mitotic exit block (*Holt et al., 2009*), our findings establish that this modification is a regulated, damage-responsive event that is conserved in human cells. We show that phosphorylation is induced by two chemically distinct DNA-damaging agents, MMS and sodium arsenite, both of which generate BER substrates. Critically, inhibition of the BER endonuclease APE1 or chemical blockade of abasic site recognition and repair abolishes pHsp70 formation. Of note, high doses and long treatment times are required to elicit Hsp70 phosphorylation with both compounds, which raises questions as to the precise driver of this response. One caveat to this lies in our bulk detection method: the use of whole-cell lysate for Western blotting seems to dilute pHsp70 signal at earlier time points, such that longer MMS treatment is required for sufficient pHsp70 to accumulate. This is illustrated by the fact that pHsp70 appears earlier in nuclear fractions than in whole-cell lysate (*Figures 3d and 4d*). While it is important to keep this limitation in mind, the dose and timing of insult still merits consideration. BER is a streamlined pathway: it is thought to follow a 'baton-passing' model, with each enzyme passing off intermediates to the next in the cascade (*Caldecott, 2020*). However, BER intermediates are cytotoxic. APE1-mediated cleavage of the DNA backbone produces single-strand breaks that, if encountered by replication forks, can be converted into DSBs (*Nikolova et al., 2010*; *Ensminger et al., 2014*). The requirement for APE1 activity, along with the involvement of the DSB repair kinases ATM and DNA-PKcs, supports a model in which BER-induced DSBs lead to Hsp70 phosphorylation. While it is true that pharmacological inhibition and siRNA-mediated knockdowns can have off-target and pleiotropic effects (*Hantschel, 2015*; *Jackson and Linsley, 2010*), our use of orthogonal approaches to perturb the aforementioned pathways in conjunction with knowledge that both MMS and sodium arsenite generate lesions repaired by BER leads us to believe that the parsimonious explanation is that phosphorylation of Hsp70 is downstream of a DDR (*Beard et al., 2019*; *Sedgwick, 2004*; *Bau et al., 2002*).

While it is clear that Hsp70 is phosphorylated in response to DNA damage, the lag time between insult and pHsp70 suggests a complex signaling pathway. Phosphorylation does not arise simply during damage exposure; instead, it requires both prolonged insult and progression through mitosis. Despite the longstanding view that MMS-induced damage primarily arises during S phase, we find

that synchronizing cells at G1/S does not enhance Hsp70 phosphorylation. This paradox may be resolved by emerging evidence that BER-induced DSBs can occur in non-replicating cells (*Polyzos et al., 2024*). Considering that the DDR is considerably rewired during mitosis, and though BER occurs, repair of lesions such as DSBs is forestalled until mitotic exit (*Heijink et al., 2013*; *Blackford and Stucki, 2020*; *Pramanik et al., 2024*), it is possible that pHsp70 arises in response to high lesion burden in M phase to prevent cell cycle entry into the vulnerable S phase.

Our genetic studies in yeast directly link phosphoregulation of Ssa1 at T492 to cell cycle progression. Both phosphonull (T492A) and phosphomimetic (T492E) mutants cause growth defects in the absence of the compensatory isoform Ssa2, underscoring the importance of dynamic, reversible phosphorylation at this site. Alternatively, it is possible that the phosphonull mutation is not neutral and thus exhibits additional effects. Future work should examine the biochemical consequences of this mutation to clarify our findings. Cell cycle profiling reveals that T492E yeast in both *SSA2* and *ssa2Δ* backgrounds accumulate in G1 and fail to efficiently enter S phase following MMS exposure, while T492A strains show delayed recovery in the absence of Ssa2. Further, in the absence of MMS insult, the Ssa1 T492E mutant strain exhibits altered growth and cell cycle progression even in the *SSA2* background, suggesting a dominant-like effect of the phosphomimetic variant. The dominant-like phenotype of T492E is consistent with biochemical data suggesting persistent, aberrant substrate engagement, akin to the effects of mutations that trap Hsp70 in a closed conformation (*Fontaine et al., 2015*). Together, these results support a model in which pHsp70 serves as a reversible molecular brake, preventing premature S phase entry under conditions of BER stress.

Interestingly, this mechanism parallels, but is distinct from the previously described phospho-Ssa1 (T36) mediated regulation of the cell cycle. Phosphorylation at T36/38 also enforces G1/S arrest in yeast. However, the residue is in the NBD, and phosphorylation decreases nucleotide binding. Furthermore, the activity is ER-localized, where it slows the accumulation of Cdk1-Cln3 by sequestering and promoting the degradation of Cln3 (*Truman et al., 2012*; *Vergés et al., 2007*). In contrast, pT492/495 occurs in the SBD, modulates ATP hydrolysis without preventing nucleotide binding, and occurs in the nucleus. Given the established role of phosphorylation in directing Hsp70-client fate (*Muller et al., 2013*; *Truman, 2017*; *Backe et al., 2025*), it is plausible that pT492/T495 controls the stability or activity of nuclear substrates critical for G1/S transition. Our data support a model wherein Hsp70 engagement with both a client and some accessory machinery (e.g. a cochaperone) are controlled by phosphorylation to regulate S phase entry. For instance, if a specific Hsp70 client must be degraded to exit G1, phosphorylation may promote engagement with said client and proper degradation machinery, and dephosphorylation could allow client release. In this case, the phosphomimetic Hsp70 would be unable to disengage, and therefore cause the G1/S stall we report. Conversely, the phosphonull Hsp70 would be unable to establish proper interactions with the client and/or accessory machinery, thereby preventing efficient degradation and leading to the observed dysregulation of S phase re-entry. While this model explains our experimental observations, other possibilities remain, and future work is required to further establish mechanistic details.

Collectively, our findings reveal a conserved phospho-switch on Hsp70 that links DNA damage sensing during BER to cell cycle control, adding an unanticipated regulatory layer to the integration of repair and checkpoint pathways. Strikingly, this pathway came to light because it is targeted by a pathogenic effector kinase, once again illustrating how pathogens can hijack, and thereby illuminate, fundamental aspects of cell biology. Such examples underscore the enduring value of pathogens as powerful tools to uncover deeply conserved regulatory mechanisms. This work opens the door to identifying the upstream kinase(s) and downstream client(s) that mediate this effect, and to determining whether T495 phosphorylation serves as a general mechanism for coordinating repair with proliferation in diverse physiological contexts. By coupling BER to a reversible molecular brake on S phase entry, phosphorylation of Hsp70 at T495 emerges as a conserved checkpoint signal that safeguards genome integrity under conditions of repair-induced stress.

During the revision of this manuscript, Omkar et al. reported heat shock-induced phosphorylation of this residue in *S. cerevisiae* via the Cell Wall Integrity (CWI) pathway (*Omkar et al., 2025*), consistent with the regulatory potential we describe here.

## Methods

### Cell culture

HEK293T and HeLa lines were obtained from ATCC. All cell lines were regularly tested for mycoplasma contamination. Cells were maintained in Dulbecco's modified Eagle's medium (Gibco) supplemented with 10% fetal bovine serum (VWR) in a humidified incubator at 37°C, 5% $CO_2$. Cells were treated with 10 mM MMS (Fisher AC156890050) for 5 hr unless otherwise indicated. Cells were treated with methoxyamine hydrochloride (Sigma-Aldrich 226904), APE1 compound III (EMD Millipore 262017), bleomycin (sulfate) (Thomas Scientific C830H18), camptothecin (Selleck S1288), hydroxyurea (Sigma-Aldrich H8627), sodium arsenite (Fisher Scientific S88733), KU-60019 (MedChem Express HY-12061), AZD1390 (MedChem Express 2089288-03-7), CCT241533 (MedChem Express HY-14715B), PF-670462 (Sigma-Aldrich SML0795), thymidine (Fisher 501882638), Ro-3306 (MedChem Express HY-12529), and AZD7648 (MedChem Express 2230820-11-6) as described in the figure legends.

### siRNA and plasmid transfection of mammalian cells

Cells were transfected with high-quality maxiprepped plasmid DNA (Sigma Genelute HP kit), custom-synthesized siRNA duplexes (Sigma-Aldrich VC30002), or Mission siRNA Universal Negative Control #1 (Sigma-Aldrich SIC001) using the jetPRIME transfection system (VWR 89129-926). For plasmid transfection, cells were grown to 70% confluence and transfected overnight with total μg of DNA added as recommended by supplier for the culture vessel. For siRNA transfection, knockdowns were performed per manufacturer suggestions in HeLa cells. Cells were transfected for 24 hr after which the media was replaced with fresh media. Cells were allowed to grow for an additional 48 hr before any additional treatment and harvesting.

### Whole-cell lysate preparation and Western blotting

Once experimental manipulations were complete, media was aspirated and cells were washed 3× with ice-cold 1× TBS on ice. Cells were gently scraped into 1 mL ice-cold 1× TBS and transferred to 1.5 mL microcentrifuge tubes. Cells were pelleted at 3000×$g$ for 5 min at 4°C, and the supernatant was aspirated. Pellets were stored at –80°C until lysis.

1× RIPA lysis buffer (Cell Signaling Technology 9806S) supplemented with 1× Roche cOmplete protease inhibitor cocktail (Roche 11697498001), 1× Roche PhosStop (Roche 04906837001), or 1× Halt Phosphatase Inhibitor Cocktail (Thermo Scientific 78440), and 1 mM PMSF was added to the frozen pellets, which were then agitated for 30 min at 4°C, then centrifuged at 16,000×$g$ for 15 min at 4°C. Cleared lysates were transferred to a new tube, and protein concentrations were quantified using the Pierce BCA Assay Kit (Thermo Scientific 23227). 30–100 μg total protein was denatured in 1× SDS sample buffer with 10 mM dithiothreitol (DTT) for 8 min at 95°C. Lysates were loaded onto polyacrylamide gels (8, 10% or Bio-Rad Mini-PROTEAN TGX Precast Protein Gels, 4–20%) (Bio-Rad) and separated by SDS-PAGE. Proteins were transferred to methanol-activated 0.2 μm pore PVDF membranes overnight at 4°C in 1× CAPS, 10% methanol. Membranes were blocked in either 5% bovine serum albumin (BSA) (VWR) for phospho-specific antibodies, or 5% non-fat dry milk (Bio-Rad) in TBS-T (0.1% Tween-20 in TBS) for 1 hr at room temperature. Primary antibodies were diluted in 5% BSA, 0.02% $NaN_3$ in TBS-T. Membranes were incubated with primary antibody solutions overnight at 4°C. Membranes were washed 3× in TBS-T for 10 min each wash, then incubated with 1:5000 secondary antibody in blocking buffer for 1 hr at room temperature. Membranes were washed 3× in TBS-T for 10 min each wash, then developed for 1 min in Amersham ECL Western Blotting Detection Reagent (Cytiva RPN2209) or Immobilon Crescendo (Millipore WBLUR0500) and imaged on a ChemiDoc Imaging System (Bio-Rad).

### Phosphatase sensitivity

HEK293T cells were treated with 10 mM MMS for 5 hr or left untreated. Following treatment, the cells were harvested and split into two 1.5 mL microcentrifuge tubes per condition. Cells were lysed in home-made RIPA buffer (20 mM Tris HCl pH 7.5, 150 mM NaCl, 1 mM EDTA, 1% Triton X-100, 1% sodium deoxycholate, 0.1% SDS) supplemented with 1 mM PMSF, 1× PhosStop (Roche 04906837001). The lysis buffer was further supplemented with 1× Roche PhosStop (Roche 04906837001) or 1× Halt Phosphatase Inhibitor Cocktail (Thermo Scientific 78440) for the '+phosphatase inhibitor' samples. Lysis, protein quantification, and Western blots were performed as described above.

## Nuclear and cytoplasmic extraction

Nuclear and cytoplasmic extraction was performed at 4°C using the NE-PER Nuclear and Cytoplasmic Reagents (Fisher Scientific PI78835, Thermo Scientific 78833) following the manufacturer's protocol. Protein quantification and Western blotting were performed as described above.

## Protein purification

Hsc70 and point mutants were purified as described previously (*Chang et al., 2010*). In brief, plasmids containing His-tag proteins were transformed into Rosetta(DE3) *Escherichia coli*. Bacteria were grown in 2 L Terrific Broth (TB) to $OD_{600}$=0.6. Bacterial cultures were cooled to 20°C, and protein expression was induced with 200 µM IPTG at 20°C overnight. The cells were harvested by centrifugation (7500 r.p.m. 10 min, JLA 8.1 rotor) and then resuspended in 20 mL His-Binding buffer (50 mM Tris, 10 mM imidazole, 500 mM NaCl, pH 8)+2 tablets EDTA-free Roche cOmplete Protease Inhibitor per liter culture using a Dounce homogenizer. Cells were then lysed by sonication at Amp 35% for 5 min, 30 s on, 30 s off, on ice at 4°C. Lysate was separated by centrifugation (18,000 r.p.m. 30 min, JA-20 Beckman rotor) and incubated with 10 mL/L pre-equilibrated Ni-NTA resin (EMD Millipore 70666-5)/L of culture at 4°C for 1 hr. Bound protein was first washed with 200 mL His-Binding buffer, then 100 mL His-Washing buffer (50 mM Tris, 30 mM imidazole, 300 mM NaCl, pH 8), and then eluted with His-Elution buffer (50 mM Tris, 300 mM imidazole, 300 mM NaCl, pH 8). This His-tag was then removed by the spiking in 5 mM β-mercaptoethanol and 600 µg TEV protease to the sample and dialyzing overnight into Buffer A (25 mM HEPES, 5 mM $MgCl_2$, 10 mM KCl, pH 7.5) in 10 KDa MWCO snakeskin dialysis tubing (Thermo Fisher 68100). The protein was further purified by an ATP-agarose column using previously established protocols (*Chang et al., 2010*). Purified DnaJA2 and tau were acquired from JEG.

## ATPase assays with MG

The ATPase activity of Hsc70 and Hsc70(T495E) was performed with MG (Sigma-Aldrich) as described previously (*Chang et al., 2008*). Briefly, in a clear 96-well plate, Hsc70 or Hsc70(T495E) were incubated with increasing concentrations of human DnaJA2 (DJA2) in 25 µL total volume. Reactions were also performed with DnaJA2 in the absence of Hsc70 for background subtraction. The assay buffer was 100 mM Tris at pH 7.4, 20 mM KCl, 6 mM $MgCl_2$, and 0.01% Triton. The reaction was initiated by the addition of ATP at a final concentration of 1 mM and incubated at 37°C for 1 hr. After incubation, 80 µL of MG reagent was added, followed by 10 µL of saturated sodium citrate to quench the reaction. Absorbance was measured at 620 nm on a SpectraMax M5 plate reader (Molecular Devices). ATP hydrolysis rates were calculated by comparison to a phosphate standard. Displayed curves are a combination of six replicates.

## FP of ATP-FAM

Nucleotide of Hsc70 and Hsc70(T495E) was assessed with an FP assay using labeled ATP. First, apo-Hsc70 was generated by subsequent 6–12 hr dialyses in the following buffers: buffer 1 (25 mM HEPES, 100 mM NaCl, 5 mM EDTA [pH 7.5]), buffer 2 (25 mM HEPES, 100 mM NaCl, 1 mM EDTA [pH 7.5]), buffer 3 (25 mM HEPES, 5 mM $MgCl_2$, 10 mM KCl [pH 7.5]). Following this, 40 kDa MWCO Zeba Buffer Exchange columns (Thermo Fisher A57760) were used to exchange the buffer to FP assay buffer (100 mM Tris, 20 mM KCl, 6 mM $MgCl_2$, pH 7.4) before protein quantification. The assay was performed in 384-well black round-bottom low-volume plates (Corning 4511). 40 nM ATP-FAM (Jena Bioscience nu-805-5fm) was added to each well for a final concentration of 20 nM in 20 µL. Hsc70 and Hsc70(T495E) were added for a starting concentration of 20 µM in 20 µL, and then serially diluted with FP assay buffer (twofold 13 times into the ATP-FAM). Plates were covered and incubated for 30 min at room temperature. Fluorescence polarization was read on a SpectraMax M5 plate reader (excitation, 485 nm; emission, 535 nm).

## Partial proteolysis

The partial proteolysis protocol to identify Hsp70 conformations was modified from previous work (*Rinaldi et al., 2018*). Hsc70 and Hsc70(T495E) were buffer-exchanged to proteolysis buffer (40 mM HEPES, 20 mM NaCl, 8 mM $MgCl_2$, 20 mM KCl, 0.3 mM EDTA, pH 8) using Zeba 40 kDa MWCO buffer exchange columns and diluted to 3 µM in this buffer. ATP or ADP was added to 1 mM and

incubated for 30 min at room temperature. 1.5 µM trypsin (Sigma EC 3.4.21.4) was added for 2 hr at room temperature. The reaction was quenched by boiling in SDS loading buffer (125 mM Tris-HCl pH 6.8, 5% SDS, 10% β-mercaptoethanol, 20% glycerol, 0.025% bromophenol blue), run on a precast SDS-PAGE gel (Bio-Rad), and stained by Coomassie.

## Hsc70-tau binding ELISA

Hsc70-tau ELISA was modified from a previously published protocol (*Nadel et al., 2024*). It was performed in a Fisherbrand, flat-bottom 96-well plate, clear, PS (cat no 12565501). 1 µM Hsc70 or Hsc70(T495E) (diluted in dialysis buffer 3 from FP assay) was added to wells along with 1 mM ATP and incubated overnight at 37°C. Protein was discarded from wells followed by 3× 3 min washes on rocker with PBS-T, discarding the PBS-T and vigorously blotting the inverted plate on a paper towel between each wash. In triplicate, add 30 µL of tau to each well, spanning a 12-dose threefold concentration gradient (from 70 µM). Plates were covered and incubated at room temperature for 3 hr. Solutions were removed and wells washed 3× with PBS-T as described above. 100 µL of blocking solution (5% non-fat milk in TBS-T) was added to all wells and incubated at room temperature for 5 min without rocking. Solution was removed without washing. 50 µL of primary antibody was added to all wells (1:2000 mouse anti-tau clone D-8 from Santa Cruz Biotechnologies in blocking solution). The plate was incubated for 1 hr at room temperature without shaking. Solution was removed and wells washed 3× with PBS-T. 50 µL of secondary antibody (1:2000 goat anti-mouse [Jackson 115-035-146] in blocking solution) was added to the wells. The plate was incubated for 1 hr at room temperature and then washed 3× with PBS-T. 50 µL of TMB substrate was added to the wells (Thermo Fisher 34028) and incubated for 15 min at room temperature in the dark. 50 µL 1 M HCl was added, and the plate read at 450 nm on SpectraMax M5 plate reader.

## Yeast strain generation

W303α yeast and W303α yeast with SSA1 NAT were gifted to us by David Morgan. Oligos for yeast point mutations were generated by PCR amplification of NAT-SSA1 with the relevant codon substituted in the primer sequence (see Key resources table). Oligos for SSA2 interruption were generated by PCR amplification of KanMX from pYM13 with flanking sequences to SSA2. gDNA was extracted from yeast by first suspending yeast colonies from a YPAD plate in 100 µL LiOAc/SDS buffer (0.2 M LiOAc, 1% SDS). Cells were incubated at 70°C for 10 min, and then 300 µL 96–100% ethanol was added. Sample was then mixed by vortexing. Sample was then spun at 15,000×*g* for 3 min, supernatant removed, and pellet dissolved in 100 µL nuclease-free water. The sample was then spun down at maximum speed for >15 s, and supernatant was transferred to a new tube. 1 µL of supernatant was used for subsequent PCRs.

For yeast transformations, competent cells were generated by first diluting an overnight culture 1:20 in YPAD and grown on a rotary shaker at 30°C to mid-log. The yeast were spun at 3000×*g* for 1 min. While spinning the yeast, 10 mg/mL salmon sperm DNA (Sigma-Aldrich D9156) was boiled for 5 min and then put on ice. The supernatant was removed from the yeast pellet, which was then washed 1× in water and re-pelleted. The water was removed and the yeast was resuspended in LiOAc/TE solution (10 mM Tris-HCl pH 8, 1 mM EDTA, 0.1 M lithium acetate [Sigma-Aldrich L4158]). This was then pelleted, the supernatant aspirated, and yeast were resuspended in 200 µL LiOAc/TE solution to generate competent cells. For each transformation, 50 µL of these competent cells were mixed with 1 µg of DNA along with 10 µL of boiled salmon sperm DNA and 500 µL of PEG/LiOAc/TE solution (10 mM Tris-HCl pH 8, 1 mM EDTA, 0.1 M lithium acetate, 40% PEG 3350). This mix was incubated at 30°C with shaking at 550 r.p.m. on a ThermoMixer F1.5 (Eppendorf), then spun for 1 min at 3000×*g*. The supernatant was removed and the yeast resuspended in sterile water and plated on YPAD and grown overnight at 30°C. The following day, these plates were replica-plated onto plates with the appropriate selection marker.

## Yeast growth

For all yeast experiments, yeast were first grown overnight in YPAD on a rotary shaker at 30°C. The following morning, yeast were diluted to OD$_{600}$=0.3 and grown at 30°C to mid-log (OD$_{600}$ = ~0.6). Strains were then concentrated (by centrifugation at 3000×*g*) to 2e7 cells/mL in 1.5 mL microcentrifuge tubes.

## Growth curve

The above yeast were diluted 1:30 into a clear flat-bottom 96-well plate (Costar 3370) and grown for 25 hr with continuous orbital shaking at 30°C on a Cytation 5 Imaging Reader (BioTek) with $OD_{600}$ collected every 20 min.

## Yeast spot test

The day prior to the experiment, Nunc Rectangular Dishes (Thermo 267060) were made with either YPAD or YPAD+0.0095% MMS. The above yeast were serially diluted 10-fold five times. 5 μL of the dilutions were spotted on the YPAD or YPAD+MMS plates. Plates were grown for 3 days at 30°C and imaged daily on a ChemiDoc Imaging System (Bio-Rad).

## Yeast cell cycle analysis

This protocol was modified from previous work (*Zhang and Siede, 2003*; *Rosebrock, 2017*). For cell cycle analysis of yeast at mid-log, we took 500 μL of the 2e7 cells/mL yeast solution, spun it at 14,000×*g* for 30 s, removed the supernatant, washed 1× in water, and re-pelleted. The water was then decanted, and the yeast were vortexed to resuspend them in the water remaining in the tube. 95% ethanol prechilled to –20°C was added to the resuspended yeast, which was then vortexed and placed on ice. For MMS treatment, yeast were grown to mid-log as previously described. 0.05% MMS (Fisher Scientific AC156890050) was then added to the yeast for 3 hr, rotating at 30°C. Samples were harvested as described for 'no recovery' conditions, and the remaining yeast was transferred to 15 mL centrifuge tubes (Fisher Scientific 12-565-268), spun at 300×*g*. The pellet was washed 1× with sterile water, spun down again, and then resuspended in the starting volume of YPAD. The caps were partially unscrewed, and the yeast was returned to the rotary at 30°C with samples harvested as detailed above at the indicated time points. Once all samples were resuspended in ethanol, the yeast were stored at –20°C for at least overnight. The yeast were then resuspended by vortexing and spun at 14,000×*g* for 1 min. The ethanol was removed, the yeast resuspended in water, and then pelleted again. The water was removed, and the pellet was resuspended in 1 mL sodium citrate buffer (50 mM sodium citrate [C8532], pH to 7.5 with citric acid [BP399]). 8 μL of 10 mg/mL RNAase A (Thermo EN0531) was then added to the samples and incubated for 2 hr at 37°C. Following this, 10 μL of 20 mg/mL Proteinase K (NEB P8107S) was added to the samples and incubated for 1 hr at 50°C. The samples were then incubated at 4°C overnight. The samples were spun down at 14,000×*g* for 1 min, and the pellets resuspended in 500 μL sodium citrate buffer+2.5 μM SYTOX Green (Thermo Fisher S7020) or sodium citrate buffer alone for the no-stain control. Samples were placed on ice and sonicated with a probe sonicator (QSonica) at 30% amplitude for 10 s. The samples were then stored in the dark at 4°C until flow cytometry analysis.

All yeast flow cytometry experiments were performed on an LSRII SORP (BD Biosciences) using a 488 nm laser, and the software analysis done on FlowJo (10.10.0). Cells were first selected by graphing FSC-A by SSC-A. Singlets were then gated by graphing FSC-A by FSC-H and then analyzed.

## Yeast lysis for Western blotting

Yeast samples were treated as described, then 1 mL of sample was pelleted in a 1.5 mL microcentrifuge tube at 13,000×*g* for 1 min, washed 1× in water, pelleted at 13,000×*g*, and then the pellets were stored at –80°C. The following day, the pellets were resuspended in 200 μL of 20% trichloroacetic acid. The samples were then centrifuged at 13,000 r.p.m. for 30 s at 4°C. The supernatant was removed, and the pellets were washed with acetone prechilled in a –20°C freezer. The samples were spun at 13,000 r.p.m. for 30 s, the supernatant removed, and the samples allowed to air-dry at room temperature for ~10 min or until fully dry. The pellets were then resuspended in 40 μL of 1 M Tris pH 7.5, and then 200 μL of 2× Laemmli Buffer+2% DTT was added to each sample. Using a 200 μL PCR tube (VWR 53509-304), ~400 μL of 0.5 mm glass disrupter beads (USA Scientific 7400-2405) were scooped into the sample, and the sample was beaten vigorously for 90 s. With a 25-gauge needle, a hole was carefully poked in the bottom of the tube. The tube was then placed in a plastic vial (Falcon 352063) and spun with tops closed at 500×*g* for 3 min. The flowthrough was transferred to a clean 1.5 mL microcentrifuge tube and boiled for 10 min. Equal volumes of sample were then run in a 8–16% precast Mini-PROTEAN TGX Stain-Free gel (Bio-Rad 456-8103) per the manufacturer's protocol. The Western blot was performed as described above.

## Data reproducibility and statistical analysis

All statistics were performed in Prism as described in the relevant figure legends. Figures are either mean ± SD or representative of the number of replicates described in the figure legend.

## Materials availability

All unique reagents generated in this study, including plasmids and yeast strains, are available from the corresponding author upon reasonable request. All data supporting the findings of this study are provided within the manuscript and supplementary files.

## Acknowledgements

We thank Dr. David Booth, Dr. David Toczyski, Dr. Adriana Steinbach, and Dr. Michael Metrick for critical evaluation and discussions of the data. We thank Dr. Henry Ng and Dr. David O Morgan for yeast strains and technical advice. We thank Dr. Oleta Johnson, Dr. Cory Nadel, and Dr. Emma Carroll from the JEG lab for technical assistance and advice. We acknowledge Vinh Nguyen for his technical support and the PFCC (RRID:SCR_018206) for assistance generating Flow Cytometry data. Research reported here was supported in part by the DRC Center Grant NIH P30 DK063720. TM acknowledges support from the MPHD T32 training grant. JEG acknowledges support from R01NS059690. SM acknowledges financial support from the National Institutes of Health (grant nos. R01GM140440 and R01GM144378), the Pew Charitable Trust (grant no. A129837), a Bowes Biomedical Investigator award, and is a Biohub, San Francisco, Investigator.

## Additional information

### Competing interests

Shaeri Mukherjee: Reviewing editor, eLife. The other authors declare that no competing interests exist.

### Funding

| Funder | Grant reference number | Author |
|---|---|---|
| National Institute of General Medical Sciences | R01GM140440 | Shaeri Mukherjee |
| National Institute of General Medical Sciences | R01GM144378 | Shaeri Mukherjee |
| National Institute of Neurological Disorders and Stroke | R01NS059690 | Jason E Gestwicki |
| Pew Charitable Trust | A129837 | Shaeri Mukherjee |
| Bowes Biomedical Investigator Program | Bowes Biomedical Investigator award | Shaeri Mukherjee |
| National Institutes of Health | MPHD T32 training grant | Thomas Moss |

The funders had no role in study design, data collection and interpretation, or the decision to submit the work for publication.

### Author contributions

Thomas Moss, Conceptualization, Data curation, Formal analysis, Validation, Investigation, Visualization, Methodology, Writing – original draft, Writing – review and editing; Alexandra Wooldredge, Koustav Bhakta, Matthew Cronin, Investigation; Jason E Gestwicki, Conceptualization, Writing – review and editing; Shaeri Mukherjee, Conceptualization, Resources, Supervision, Funding acquisition, Writing – original draft, Writing – review and editing

**Author ORCIDs**
Thomas Moss ⬤ https://orcid.org/0000-0003-3647-9319
Alexandra Wooldredge ⬤ https://orcid.org/0009-0007-4636-0342
Koustav Bhakta ⬤ https://orcid.org/0000-0002-2180-015X
Matthew Cronin ⬤ https://orcid.org/0009-0004-3524-6227
Shaeri Mukherjee ⬤ https://orcid.org/0000-0003-3820-0174

Reviewer #1 (Public review): https://doi.org/10.7554/eLife.110044.3.sa1
Reviewer #2 (Public review): https://doi.org/10.7554/eLife.110044.3.sa2
Reviewer #3 (Public review): https://doi.org/10.7554/eLife.110044.3.sa3
Author response https://doi.org/10.7554/eLife.110044.3.sa4

## Additional files

### Supplementary files
MDAR checklist

### Data availability
All data supporting the findings of this study are provided within the manuscript and supplementary files. Source data files have been provided.

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

# Appendix 1

**Appendix 1—key resources table**

| Reagent type (species) or resource | Designation | Source or reference | Identifiers | Additional information |
|---|---|---|---|---|
| Transfected construct (human) | Hsc70 in pMCSG7 | Moss 2019 *Seo et al., 2016* | pSM226 | |
| Transfected construct (human) | Hsc70(T495E) in pMCSG7 | Moss 2019 *Seo et al., 2016* | pSM227 | |
| Transfected construct (human) | LegK4Δ1–58 in pEGFP-C2 | Moss 2019 *Seo et al., 2016* | pSM160 | |
| Transfected construct (human) | MPG in pIRESNeo | Addgene | Plasmid #12548 | |
| Transfected construct (human) | APE1 in pCMVd3 | This paper | pSM303 | Plasmid for APE1 overexpression experiment in human cells (see *Figure 2—figure supplement 1*) |
| Transfected construct (human) | pCMVd3 empty vector | This paper | pSM304 | Empty vector control for APE1 overexpression experiment in human cells (see *Figure 2—figure supplement 1*) |
| Transfected construct (human) | Polβ in pLVX-GWE-IRES-puro | Addgene | Plasmid #128653 | |
| Transfected construct (human) | plVX-GWE-IRES-puro empty vector | This paper | pSM289 | Empty vector control for Polβ overexpression experiment in human cells (see *Figure 2—figure supplement 1*) |
| Transfected construct (human) | pYM13 PCR template KanMX resistance | Gift from David O. Morgan Lab | pSM302 | |
| Transfected construct (human) | siRNA to DNA-PKcs #1 | genomeRNAi | s773 | GCGUUGGAGUGCUACAACA[dT][dT] |
| Transfected construct (human) | siRNA to DNA-PKcs #2 | genomeRNAi | s775 | CAAGCGACUUUAUAGCCUU[dT][dT] |
| Transfected construct (human) | siRNA to DNA-PKcs #3 | genomeRNAi | SI02663633 | GACCCUGUUGACAGUACUU[dT][dT] |
| Transfected construct (human) | siRNA to ATM #1 | genomeRNAi | s1710 | GCUGUUACCUGUUUGAAAA[dT][dT] |
| Transfected construct (human) | siRNA to ATM #2 | genomeRNAi | SI02663360 | CACCUGUUUGUUAGUUUAU[dT][dT] |
| Transfected construct (human) | siRNA to ATM #3 | genomeRNAi | SI03068506 | CAGCUGUCAUCAUAUAAGA[dT][dT] |
| Transfected construct (human) | siRNA to ATR #1 | genomeRNAi | s534 | GAGCCGAUUUUUAAGUCAA[dT][dT] |
| Transfected construct (human) | siRNA to ATR #2 | genomeRNAi | s535 | GAUGAGUAUGCAAAAUUUA[dT][dT] |
| Transfected construct (human) | siRNA to ATR #3 | genomeRNAi | SI02625476 | GCCGCUAAUCUUCUAACAU[dT][dT] |
| Sequence-based reagent | SSA1 gDNA T492E Forward | This paper | PCR primers | TCGATGTCGACTCTAACGGTATTTTGAATGTTTCC GCCGTCGAAAAGGGTGAAGGTAAGTCTAACAAG |
| Sequence-based reagent | SSA1 gDNA T492A Forward | This paper | PCR primers | TCGATGTCGACTCTAACGGTATTTTGAATGTTTCC GCCGTCGAAAAGGGTGCTGGTAAGTCTAACAAG |
| Sequence-based reagent | SSA1 gDNA Reverse | This paper | PCR primers | CAGATCATTAAAAGACATTTTCGTTATTATCAATTG CC GCACCAATTGGCGCATGCCGGTAGAGG |
| Sequence-based reagent | SSA2-KanMx Forward | This paper | PCR primers | TTGATTAATTCCAACAGATCAAGCAGATTTTATACA GAAATATTTATACAATGGGTAAGGAAAAGACTCA |

*Appendix 1 Continued on next page*

*Appendix 1 Continued*

| Reagent type (species) or resource | Designation | Source or reference | Identifiers | Additional information |
|---|---|---|---|---|
| Sequence-based reagent | KanMx-Ssa2 Reverse | This paper | PCR primers | GGAAAGCAAAAGTAAAACTTTTCGGATATTTTACAGG GCGATCGCTAAGCTTAGAAAAACTCATCGAGCA |
| Antibody | Anti-pHsp70 (mouse, monoclonal) | Custom (Moss 2019) *Seo et al., 2016* | | WB (1:5000) |
| Antibody | Anti-GAPDH (mouse, monoclonal) | Proteintech | 60004-1-Ig | WB (1:3000) |
| Antibody | Anti-γH2AX (mouse, monoclonal) | Millipore Sigma | 05-636 | WB (1:1000) |
| Antibody | Anti-Hsp70 (mouse, monoclonal) | Santa Cruz Biotechnology | sc-66048 | WB (1:1000) |
| Antibody | Anti-phospho-DNA-PKcs (S2056) (rabbit, polyclonal) | Abcam | ab18192 | WB (1:1000) |
| Antibody | Anti-phospho-Chk2 (T68) (rabbit, monoclonal) | Cell Signaling Technology | 2197T | WB (1:1000) |
| Antibody | Anti-Hsp90 (rabbit, polyclonal) | Cell Signaling Technology | 4874S | WB (1:1000) |
| Antibody | Anti-phospho-Chk2 (S516) (rabbit, polyclonal) | Cell Signaling Technology | 2669T | WB (1:1000) |
| Antibody | Anti-phospho-ATM (S1981) (mouse, monoclonal) | Cell Signaling Technology | 4526S | WB (1:1000) |
| Antibody | Anti-tubulin (mouse, monoclonal) | Proteintech | 66031-1-Ig | WB (1:3000) |
| Antibody | Anti-thymidine kinase (rabbit, monoclonal) | Cell Signaling Technology | 28755S | WB (1:1000) |
| Antibody | Anti-phospho-CDK1(Y15) (rabbit, monoclonal) | Cell Signaling Technology | 4539S | WB (1:1000) |
| Antibody | Anti-cyclin B (mouse, monoclonal) | Santa Cruz Biotechnology | sc-245 | WB (1:500) |
| Antibody | Anti-phospho-histone-H3 (S10) (rabbit, monoclonal) | Cell Signaling Technology | 53348S | WB (1:1000) |
| Antibody | Anti-cyclin E (mouse, monoclonal) | Santa Cruz Biotechnology | sc-247 | WB (1:1000) |
| Antibody | Anti-CDT1 (rabbit, monoclonal) | Cell Signaling Technology | 8064S | WB (1:1000) |
| Antibody | Anti-lamin B1 (rabbit, polyclonal) | Proteintech | 12987-1-AP | WB (1:1000) |
| Antibody | Anti-cyclin A (mouse, monoclonal) | Santa Cruz Biotechnology | sc-271682 | WB (1:1000) |
| Antibody | Anti-Polβ (rabbit, polyclonal) | Proteintech | 18003-1-AP | WB (1:1000) |
| Antibody | Anti-APE1 (rabbit, polyclonal) | Proteintech | 10203-1-AP | WB (1:1000) |

*Appendix 1 Continued on next page*

*Appendix 1 Continued*

| Reagent type (species) or resource | Designation | Source or reference | Identifiers | Additional information |
|---|---|---|---|---|
| Antibody | Anti-ATM (rabbit, monoclonal) | Cell Signaling Technology | 2873S | WB (1:1000) |
| Antibody | Anti-ATR (rabbit, polyclonal) | Cell Signaling Technology | 2790S | WB (1:1000) |
| Strain, strain background (*S. cerevisiae*) | W303α | Gift from David O. Morgan Lab | Y017 | Parental strain for yeast mutants |
| Strain, strain background (*S. cerevisiae*) | Ssa1(WT)-NAT #1 | Gift from David O. Morgan Lab | Y019 | *S. cerevisiae* strain with WT Ssa1 (see ***Figure 5***) |
| Strain, strain background (*S. cerevisiae*) | Ssa1(WT)-NAT #2 | This paper | Y034 | Independently generated *S. cerevisiae* strain with WT Ssa1 (see ***Figure 5***) |
| Strain, strain background (*S. cerevisiae*) | Ssa1(T492A)-NAT #1 | This paper | Y040 | Independently generated *S. cerevisiae* strain with Ssa1 T492A (see ***Figure 5***) |
| Strain, strain background (*S. cerevisiae*) | Ssa1(T492A)-NAT #2 | This paper | Y041 | Independently generated *S. cerevisiae* strain with Ssa1 T492A (see ***Figure 5***) |
| Strain, strain background (*S. cerevisiae*) | Ssa1(T492E)-NAT #1 | This paper | Y038 | Independently generated *S. cerevisiae* strain with Ssa1 T492E (see ***Figure 5***) |
| Strain, strain background (*S. cerevisiae*) | Ssa1(T492E)-NAT #2 | This paper | Y039 | Independently generated *S. cerevisiae* strain with Ssa1 T492E (see ***Figure 5***) |
| Strain, strain background (*S. cerevisiae*) | Ssa1(WT)-NAT;*ssa2Δ* #1 | This paper | Y023 | Independently generated *S. cerevisiae* strain with WT Ssa1 and disrupted SSA2 (see ***Figure 5***) |
| Strain, strain background (*S. cerevisiae*) | Ssa1(WT)-NAT;*ssa2Δ* #2 | This paper | Y037 | Independently generated *S. cerevisiae* strain with WT Ssa1 and disrupted SSA2 (see ***Figure 5***) |
| Strain, strain background (*S. cerevisiae*) | Ssa1(T492A)-NAT;*ssa2Δ* #1 | This paper | Y051 | Independently generated *S. cerevisiae* strain with Ssa1 T492A and disrupted SSA2 (see ***Figure 5***) |
| Strain, strain background (*S. cerevisiae*) | Ssa1(T492A)-NAT;*ssa2Δ* #2 | This paper | Y046 | Independently generated *S. cerevisiae* strain with Ssa1 T492A and disrupted SSA2 (see ***Figure 5***) |
| Strain, strain background (*S. cerevisiae*) | Ssa1(T492E)-NAT;*ssa2Δ* #1 | This paper | Y044 | Independently generated *S. cerevisiae* strain with Ssa1 T492E and disrupted SSA2 (see ***Figure 5***) |
| Strain, strain background (*S. cerevisiae*) | Ssa1(T492E)-NAT;*ssa2Δ* #2 | This paper | Y057 | Independently generated *S. cerevisiae* strain with Ssa1 T492E and disrupted SSA2 (see ***Figure 5***) |
| Cell line (*Homo sapiens*) | HeLa | ATCC | CRM-CCL-2 | |
| Cell line (*H. sapiens*) | HEK293T | ATCC | CRL-3216 | |
| Software | Prism Version 10.4.2 | GraphPad | | https://www.graphpad.com/scientific-software/prism/ |
| Software | ImageLab 6.0.1 | Bio-Rad | | https://www.bio-rad.com/en-us/product/image-lab-software?ID=KRE6P5E8Z |
| Software | FlowJo 10.10.0 | BD Life Sciences | | https://flowjo.com/flowjo/download |

*Appendix 1 Continued on next page*

*Appendix 1 Continued*

| Reagent type (species) or resource | Designation | Source or reference | Identifiers | Additional information |
|---|---|---|---|---|
| Software | UCSF Chimera X v1.3 | Pettersen et al. ***Blackford and Stucki, 2020*** | | https://www.rbvi.ucsf.edu/chimera |
| Chemical compound, drug | Methyl methanesulfonate | Fisher Scientific | AC156890050 | |
| Chemical compound, drug | Methoxyamine hydrochloride | Sigma-Aldrich | 226904 | |
| Chemical compound, drug | APE1 compound III | EMD Millipore | 262017 | |
| Chemical compound, drug | Bleomycin (sulfate) | Thomas Scientific | C830H18 | |
| Chemical compound, drug | Camptothecin | Selleck | S1288 | |
| Chemical compound, drug | Hydroxyurea | Sigma-Aldrich | H8627 | |
| Chemical compound, drug | Sodium arsenite | Fisher Scientific | S88733 | |
| Chemical compound, drug | KU-60019 | MedChem Express | HY-12061 | |
| Chemical compound, drug | AZD1390 | MedChem Express | 2089288-03-7 | |
| Chemical compound, drug | CCT241533 | MedChem Express | HY-14715B | |
| Chemical compound, drug | PF-670462 | Sigma-Aldrich | SML0795 | |
| Chemical compound, drug | Thymidine | Fisher Scientific | 501882638 | |
| Chemical compound, drug | Ro-3306 | MedChem Express | HY-12529 | |
| Chemical compound, drug | AZD7648 | MedChem Express | 2230820-11-6 | |

