## [Editor Report · eLife Assessment]

This potentially **valuable** manuscript focuses on the phosphorylation of residue T495 as a mechanism to inactivate HSP70 and disrupt cell cycle progression in response to DNA damage. The evidence supporting this model is **solid**, but would be significantly strengthened by additional studies defining the extent of T495 phosphorylation induced by DNA damage, identifying the kinase responsible for phosphorylating T495 of HSP70, and further elucidation of the functional implications of T495 phosphorylation in human cells. This work will be of interest to scientists focused on topics including chaperone biology, proteostasis, cell cycle progression, and DNA damage.

---

## [Referee Report · Reviewer #1 (Public review)]

Summary:

This study identifies a conserved phosphorylation event on Hsp70, at human T495 that is triggered by DNA damage. The authors show that this modification arises in response to MMS and is temporally associated with cell cycle progression through mitosis. Using biochemical analysis, they further argue that the phosphomimetic Hsc70(T495E) adopts an open-like conformation with impaired J protein-stimulated ATP hydrolysis while still retaining client binding. In yeast, both phosphomimetic and phosphonull mutants perturb growth and cell cycle progression, supporting the idea that dynamic regulation of this site helps coordinate DNA damage responses with G1/S control.

Strengths:

A major strength of the paper is that it links prior work on Legionella-mediated Hsp70 phosphorylation to a normal cellular DNA damage response. The study is also commendably multi-level, combining mammalian cell biology, in vitro biochemistry, and yeast genetics to support the central model. Together, the authors provide a coherent story that this Hsp70 site has functional importance in checkpoint-like control rather than being a passive phosphosite, adding to our understanding of the chaperone code.

Minor Weaknesses:

The authors acknowledge that the direct kinases/phosphatases for this site remain unknown. Some conclusions are therefore still somewhat inferential, especially the model that pHsp70 acts as a reversible molecular brake on S-phase entry. These limitations do not undermine the importance of these exciting findings, but they do leave the paper somewhat short of a fully resolved mechanism.

Comments on revisions:

The authors have done a great job in addressing all the previous reviewer concerns. They have provided additional data and refined the text, stating limitations of their proposed model. In doing so, they have produced a much-improved version of the manuscript.

---

## [Referee Report · Reviewer #2 (Public review)]

The revised manuscript offers little new information and fails to address the critical weaknesses identified in the original submission.

While we can agree that phosphorylation of Thr495 would likely affect Hsp70 function-given the known biochemistry of Hsp70s and the author's previous work on LegK4-the significance of this finding hinges on whether it is a regulated process. If a meaningful fraction of Hsp70 were phosphorylated in a regulated manner triggered by DNA damage or cell cycle progression, it would constitute an important discovery, regardless of its specific impact on fitness in a given context.

However, beyond highlighting the temporal profile of Hsp70 phosphorylation in MMS-treated cells (Figure 4e), the paper fails to rule out the possibility that this correlation is merely an irrelevant side reaction. This "bystander" phosphorylation could simply be caused by the activation of kinases during the experimental MMS treatment and subsequent washout. The authors' claim-that the fraction of phosphorylated Hsp70 increases in a "regulated, cell-cycle dependent manner"-does not sufficiently counter the possibility of it being a non-functional side effect.

This concern could be resolved if the authors had identified the specific kinase, demonstrated its specificity, and manipulated it either genetically or pharmacologically. While I acknowledge this is a "tall order," the lack of such data limits the paper's significance. Furthermore, the current data fails to meet a much lower bar: confirming that a substantial fraction of Hsp70 is actually phosphorylated under the tested conditions. Such a finding would at least suggest the event is capable of impacting the overall Hsp70 pool.

It is surprising that the authors have not provided a ratiometric assay to settle this, such as an immunoblot of total Hsp70 separated on a Phos-tag or IEF gel. Instead, they rely on indirect evidence and data subject to alternative interpretations. Specifically, they argue that the fitness cost of the Thr495Ala mutation (or the phosphomimetic mutation) is due to the loss of regulatory phosphorylation (or deregulated phosphorylation); however, it is equally plausible that the mutations create Hsp70 hypomorphs whose defects are only exposed under stressful experimental conditions.

---

## [Referee Report · Reviewer #3 (Public review)]

In this manuscript Moss et al. demonstrate that Hsp70 phosphorylation at a conserved threonine residue integrates DNA damage responses with cell-cycle control. The authors present unbiased biochemical, cell-based, and yeast genetic analyses showing that phosphorylation of human Hsp70 at T495 (and the analogous Ssa1 T492 in yeast) is triggered by base-excision-repair intermediates and downstream DDR kinase activity, leading to delayed G1/S progression after DNA damage. They used orthogonal approaches such as ATPase assays, phospho-specific detection, kinase-inhibition studies, synchronization experiments, and phenotypic analyses of phosphomutants. They presented robust data which collectively supported the conclusion that dynamic Hsp70 phosphorylation functions as a conserved "molecular brake" to prevent inappropriate S-phase entry under genotoxic stress.

Comments on revisions:

The authors have addressed all my questions and concerns.

---

## [Author Response]

The following is the authors’ response to the original reviews

We thank the reviewers for their time and consideration of the manuscript. We have added new data to Figure 5 (Figure 5a) to address concerns regarding the conservation of the Hsp70 phosphorylation in yeast. Additionally, we have changed the title of the manuscript to “Hsp70 is phosphorylated in a conserved response to DNA damage and contributes to cell cycle control” to more accurately represent the conclusions we draw.

**Public Reviews:**

**Reviewer #1 (Public review):**
The strength of evidence of the mechanistic and "conserved checkpoint" claims that this site is directly activated by DNA damage is inadequate and fundamentally incorrect.

We respectfully disagree with the reviewer’s characterization of our conclusions. Our data demonstrate that DNA damage induces this phosphorylation in a cell-cycle–dependent manner. We do not claim to have defined the direct kinase or full mechanistic pathway; rather, we establish that site activation is damage-responsive and functionally linked to cell-cycle regulation. Consistent with this, phospho-mutants in yeast exhibit clear cell-cycle defects, supporting a conserved functional role. We address each of the reviewer’s specific concerns below.

Specific comments:(1) Activation of T495:The author's premise for the site being activated by DNA damage is Albuquerque et al, where PTMs on MMS treated yeast are analyzed. T492 (the yeast equivalent of human T495) is observed as phosphorylated. However, the authors fail to note that there is no untreated sample analysis in this study, and it is likely that T492 phosphorylation is also present in untreated cells. This is also backed up by later evidence from the same lab (Smolka et al), where they do not identify T492 as being dependent on Mec1/Tel/Rad53 kinases.

We agree with this assessment of the Albuquerque study. Accordingly, we used their data to generate the hypothesis that this site is phosphorylated, and we took it upon ourselves to more rigorously demonstrate phosphorylation with appropriate controls. The validated antibody that we had previously generated[1] to track pHsp70 was the enabling technology to directly track this phosphorylation event. We now directly show phosphorylation of this site (Figure 5a, lines 276-284). Of note, as Reviewer 1 suggested, there is a smaller amount of pHsp70 in the untreated cells, which corresponds with findings from Holt et al 2009 [2]. This could reflect a baseline role of Hsp70 phosphorylation for normal growth that is accentuated upon MMS insult.

(2) The kinase(s) directly responsible for T495 phosphorylation are not identified. Instead, the authors show that knockdown or pharmacological inhibition of DNA-PKcs, ATM, Chk2, and CK1 attenuate pHsp70.

We agree with reviewer 1 that identifying the direct kinase would be an exciting finding, and we believe our manuscript will provide the foundation for future studies to address these questions. While these findings will be impactful, we do not believe their lack detracts from the observations we have made.

(3) ATM siRNA knockdown has no effect, while ATM inhibitors do, which the authors acknowledge but do not resolve. This discrepancy raises concerns about off-target drug effects.

We agree with reviewer 1 that off-target drug effects are always a concern when employing pharmacological inhibitors. To that end, we tested structurally distinct inhibitors of ATM (Figure 3b) to decrease the likelihood of the same off target effect. While complementing this with a genetic knockdown would be ideal, the discrepancies between pharmacological and genetic inhibition of ATM have been well reported (lines 214-216).[3,4] Parallel discrepancies in other kinases have been mechanistically explored by other groups.[5] The preponderance of pharmacological evidence in conjunction with RNAi suggests the most likely interpretation of our data is that ATM is involved in signaling upstream of Hsp70 phosphorylation. Thus, our data compel future work to use more sophisticated genetic methods to more specifically determine how ATM connects with pHsc70.

(4) No in vitro kinase assays, motif analysis, or phosphosite mapping confirming these kinases as direct T495 kinases are presented. Thus, the proposed signaling cascade remains speculative.

We agree that we should carefully circumscribe our conclusions about the potential signaling cascade. To communicate our conclusions more clearly, we rewrote lines 223-226 to highlight that our findings implicate these kinases in upstream signaling rather than direct phosphorylation of Hsp70.

(5) Smolka and many other labs characterized DDR sites as SQ/TQ motifs, and T492 doesn't fit that motif.

We agree, and our response to comment 4 addresses this point. Briefly, we do not claim that Hsp70 is a direct target for DDR. Notably, the SQ/TQ motifs mentioned specifically pertain to ATM and DNA-PK[6], though we would like to note several studies have demonstrated DNA-PK phosphorylation outside of these motifs.[7] Chk2 and CK1 do not prefer SQ/TQ motifs[9]. Additionally, Chk2 is known to phosphorylate non-consensus sequences as well[10].

(6) No genetic tests in yeast (e.g., BER mutants) are used to connect Ssa1 T492 phosphorylation to BER in that system, despite the strong BER-centric model.

We agree that it would be interesting to study BER mutants in yeast, and we believe this will be an exciting prospect for future studies to better establish the signaling cascade. We have included a Western blot (Figure 5a) showing that MMS treatment causes increased Hsp70 phosphorylation in yeast. MMS damage is repaired through BER in *S. cerevisiae*,[11] and the pathway itself is highly conserved.[12] Our experiments demonstrate that the phosphorylation of Hsp70 occurs as a conserved response to alkylation damage, which is the major conclusion of our paper.

(7) Overexpression of MPG gives only a modest increase in pHsp70, while APE1 overexpression has no effect, and Polβ overexpression does not decrease pHsp70. These mixed results weaken the central claim that Hsp70 phosphorylation is a tuned sensor of BER burden.

We appreciate this incisive question. Though not immediately intuitive, we do not believe these results are necessarily ‘mixed’. The lack of APE1 over-expression having an effect could be attributed to APE1 activity being necessary for the phosphorylation, but not rate-limiting. Regarding Polβ, it is important to note that not its binding, but rather its dRP lyase activity is rate-limiting in base excision repair.[13] As such, if binding sites are already saturated or near saturated, but the lyase activity remains slow, we may not observe a decrease in BER intermediates. While we do claim that phosphorylation of Hsp70 is triggered by BER intermediates (lines 193-194), we do not claim that pHsp70 is a tuned sensor of BER burden.

(8) A major concern is that pHsp70 is only convincingly detected after very high, prolonged MMS (10 mM, 5 h) or 0.5 mM arsenite treatments. Other DNA-damaging agents (bleomycin, camptothecin, hydroxyurea) that robustly activate DDR kinases do not induce pHsp70. This suggests to me that the authors are observing a side effect of proteotoxic stress. This is likely (see Paull et al, PMID: 34116476).

Our data indicate that pHsp70 specifically occurs downstream of base excision repair. Therefore, it is not surprising that drugs that do not activate BER (bleomycin, camptothecin, hydroxyurea) do not elicit the same response. While pHsp70 may arise due to DSBs generated through BER, the fact we do not see phosphorylation after bleomycin treatment could be explained by the cell-cycle dependencies we report (Figure 4e). It is also important to note that MMS-induced pHsp70 occurs primarily in the nucleus, and Western blots of whole cell lysate will contain large amounts of cytosolic Hsp70 that could dilute the signal. Indeed, in our nuclear extraction (Figure 4d), we see faint pHsp70 signal as soon as 1 h after treatment, though it increases in robustness as the time-course progresses. These data are both concordant with a model in which high BER-induced lesion burden in mitosis leads to Hsp70 phosphorylation in late M/G1.

We would like to add that, in the review article cited by Reviewer 1, the authors specifically cite studies implicating a loss-of-function in DDR pathways leading to increased proteotoxic stress (e.g. ATM deficient cells producing higher levels of aggregated proteins compared to WT). However, we find that inhibition of DDR kinases decreases, rather than increases Hsp70 phosphorylation. We thus believe that DNA damage rather than proteotoxic stress is the parsimonious cause of Hsp70 phosphorylation.

(9) A recent study in Nature Communications (Omkar et al., 2025) demonstrates rapid phosphorylation of yeast T492 in a pkc1-dependent manner, diminishing the impact of these findings.

We were excited to see this paper when it was published 3 months after we posted a preprint on bioRxiv, which was released three weeks after our submission to eLife. Rather than diminishing the impact of this paper, we believe that independent lines of evidence from different groups mutually reinforces the impact of the work. We have added a sentence to say that during the review of our work, this group independently observed this phosphorylation event in response to a different stress (lines 421-423). We believe in celebrating the scientific process arriving at consistent results, and the editorial policies of eLife reinforce that philosophy by offering ‘scoop protection.’

We would also like to highlight several differences between the scope of our papers. The phosphorylation reported by Omkar et al. appears highly constrained to yeast as part of the Cell Wall Integrity pathway, whereas ours occurs as a more highly conserved response. Additionally, our paper provides additional biochemical insight into the consequences of this phosphorylation, which is lacking in Omkar et al. If anything, this paper highlights the important regulatory capacity of this residue on Hsp70, and suggests it may serve multiple functions in the cell.

(2) Downstream Effects of T492/T495:(10) The manuscript's central conceptual advance is that pHsp70 is a cell-cycle-regulated brake on G1/S. Yet in mammalian cells, the authors show only that pHsp70 appears late, after cells have traversed mitosis, and that blocking CDK1 (G2/M) prevents its accumulation.

We would like to clarify the central contribution of this study. Prior work identified this phosphorylation in yeast, but its existence and conservation in human cells had not been established. A primary advance of our study is demonstrating that this site is phosphorylated in mammalian cells and that its accumulation is cell-cycle regulated — coinciding with late M/G1.

We further show that phosphorylation depends on cell-cycle progression, as CDK1 inhibition prevents its accumulation. While these data establish regulation, we agree that they do not by themselves define causality in mammalian cells. To address functional consequences, we leveraged the genetic tractability of *S. cerevisiae*. Phosphomimetic Ssa1 T492E increases the proportion of G1 cells in the absence of MMS and enforces a stronger G1 arrest following MMS treatment. Together, these findings support a conserved, cell-cycle–linked role for this phosphorylation and provide a foundation for future mechanistic work in mammalian systems.

(11) There is no functional test in human cells: no knockdown/rescue experiments with T495A or T495E, no cell-cycle profiling upon altering Hsp70 phosphorylation state, and no demonstration that pHsp70 actually causes any delay in S-phase entry, rather than simply correlating with late damage responses. The strong conclusion that pT495 "stalls cell cycle progression" (e.g., Figure 6 model) is therefore not supported in the human system.

We agree that we did not directly test the functional consequences of Hsp70 phosphorylation in human cells. Our intent was not to claim that we have demonstrated causality in the mammalian system, but rather to establish that this conserved phosphorylation exists in human cells and is cell-cycle regulated.

We instead used *S. cerevisiae* to interrogate this due to its increased genetic tractability. In this system, phosphomimetic mutation increases the proportion of G1 cells under basal conditions and enhances G1 arrest following MMS treatment, mirroring the damage-associated phenotype observed in human cells. These findings support a conserved functional role for this modification, although we agree that direct mechanistic testing in mammalian cells will be important for future work.

While we intended the cartoon model to be a speculative illustration of what may be occurring in order to motivate future studies. We now see how this may lead to confusion, so to improve clarity, we have removed Figure 6 from the manuscript.

(12) All functional conclusions rely on T492A/E point mutants at the endogenous SSA1 locus, usually in an ssa2Δ background, in a family of highly redundant Hsp70s. Without showing that this site is actually modified during their MMS treatments, the assignment of phenotypes to loss of a physiological phospho-switch is premature. The authors need to repeat their studies in an Ssa1-4 background, as in https://pubmed.ncbi.nlm.nih.gov/32205407/.

Thank you for this feedback. We have included a Western blot to Figure 5 (Figure 5a) addressing this comment. Briefly, we show that, in yeast, Hsp70 phosphorylation increases upon MMS treatment and is not detectable in the point-mutants in the *ssa2∆* background. The latter data suggest that Ssa3-4 modification is negligible in our system.

(13) The authors infer that T495E "locks" Hsc70 in a pseudo-open state based on reduced J-protein-stimulated ATPase activity, unchanged ATP binding, altered trypsin sensitivity, and retained tau binding. However, there is no direct comparison of phosphorylated vs T495E protein (e.g., via in vitro phosphorylation with LegK4 followed by side-by-side biochemical assays, or structural analysis). Thus, it remains unclear to what extent the glutamate substitution mimics a phosphate at this position.

Previously we did show that phosphorylation impacts the ATPase cycle of Hsp70.[1] In this paper, with the phosphomimetic mutant we see an even greater decrease of activity. This is consistent with incomplete phosphorylation yielded by in vitro phosphorylation with LegK4.[1] Due to this incomplete phosphorylation in vitro, we determined that the phosphomimetic mutant would be more useful for the assays we performed, as they rely on bulk readouts.

(14) No client release kinetics, co-chaperone binding assays, or in vivo chaperone function tests are provided, yet the discussion builds a detailed model of a "pseudo-open" state that simultaneously resembles ATP-bound conformation and allows persistent substrate engagement.

We have shown that the conformational cycle of Hsp70 (T495E) is uncoupled from nucleotide state, and that the overall conformation resembles ATP-bound Hsp70. This is consistent with prior studies on AMPylation of the same residue.[14] Additionally, we demonstrate that substrate engagement is similar between WT and T495E. This is consistent with our previously published work showing increased pHsp70 on polysomes,[1] as well as our observations that the phosphomimetic mutant in yeast exerts a phenotype even in the presence of the compensatory isoform SSA2. This dominant-like phenotype is consistent with those seen in mutations locking Hsp70 in a ‘closed’ conformation.[15] We agree that future studies examining client release kinetics and co-chaperone binding would be useful for future structural studies validating and elaborating on our findings.

**Reviewer #2 (Public review):**
Weaknesses:The kinase(s) responsible for the phosphorylation have not been identified (and hence remain inaccessible to experimental i.e., genetic or pharmacological manipulation). The mechanistic links to DNA damage repair and the fitness benefits of this proposed adaptation remain obscure. Of greater concern, the data provided in the paper fail to exclude the trivial possibility that the phosphorylation event described (and characterized through biochemical proxies) is biologically neutral, reflecting nothing more than a bystander event in which kinase(s) activated by application of high concentrations of a powerful alkylating agent (MMS) phosphorylate, at meaninglessly low stoichiometry, an abundant protein (Hsp70) on a surface exposed residue. Failure to exclude this (plausible) scenario is this paper's weakness.

We agree that we have not directly quantified the absolute stoichiometry of Hsp70 phosphorylation. However, several lines of evidence argue against the interpretation that this represents a biologically neutral, bystander modification.

First, our pulse-chase experiment (Figure 4e) shows that, after MMS removal, pHsp70 levels increase as cells progress through the cell cycle. Notably, total Hsp70 levels remain constant. This indicates that the fraction of phosphorylated Hsp70 increases in a regulated, cell-cycle dependent manner, rather than through a bystander event during acute stress.

Second, functional perturbation of the homologous site in yeast produces phenotypic consequences. The phosphomimetic Ssa1(T492E) mutant exhibits reduced growth, increased G1 accumulation, and impaired cell-cycle re-entry following MMS treatment (Figure 5). These phenotypes argue that the modification of this residue is functionally consequential.

While the upstream kinase remains to be identified, the genetic and cell-cycle phenotypes observed upon site perturbation argue that this phosphorylation is functionally consequential.

**Reviewer #2 (Recommendations for the authors):**
(1) The biochemical characterization of the phosphomimetic mutation (T495E) is thorough, relying on ATPase assays and conformational analysis. Figure 1b demonstrates reduced J-protein-stimulated ATPase activity, and Figure 1d shows an ATP-like proteolysis pattern consistent with an open conformation. As the authors are well aware, Hsp70 chaperones act on their substrates via a dynamic cycle that includes binding, ATP hydrolysis, and conformational shifts. One wonders, therefore, at the relevance of the measurement shown in Figure 1f. While it is highly plausible that the T495E mutation mimics the phosphorylation event (BiP T518E mimics key aspects of AMPylation), the lack of a biochemical characterisation of Hsp70 with pThr495 is an important limitation of this paper. Even if such a preparation cannot be accomplished with the endogenous kinase(s) whose identity remains unknown, a characterisation of LegK4-phosphorylated Hsp70 should suffice.

We agree with Reviewer 2 that the rationale for figure 1f does not logically follow the results of 1b and 1d. Rather, this experiment was motivated by the prior findings that phosphorylation of Hsp70 by *L.p.* lead to an increase occupancy on polysomes[1] (lines 137-139). We sought to better understand the discrepancy between this finding and our own by assaying the capacity of the T495E mutant to bind substrate.

Reviewer 2 raises a valid point in that phosphomimetic proteins do not necessarily behave the same as truly phosphorylated proteins. Previous work from our lab characterized the ATPase activity and in vitro folding capacity of Hsc70 that had been directly phosphorylated by LegK4[1] (lines 114-115). We were motivated to turn to a phosphomimetic mutant as LegK4 only phosphorylates around half of the Hsc70 present in solution[1] (line 116); this mixture of species makes batch analysis difficult. As we had previously published with the in vitro phosphorylated Hsc70, we didn’t believe it necessary to include along with our future analyses.

(2) As noted, the kinase(s) that phosphorylate T495 remain to be identified and is inaccessible genetically. The phenotypic consequences of impaired pThr495 are therefore assessed by a T495A mutation. This most certainly eliminates phosphorylation at that site however, Figure 5C shows quite clearly that the T/A mutation is not neutral. This is expected, given the role of an H-bond network centered upon the homologous residue in the ADP-bound configuration of Hsp70's. Importantly, the biochemical non-neutrality of the T/A mutation also compromises the interpretation of the associated phenotype, as this cannot be attributed solely to a loss of phosphorylation; it may reflect features of the T/A mutations exposed by MMS, but unrelated to the inability of the residue to undergo regulated phosphorylation.

We appreciate this insightful critique. We agree that the alanine substitution may perturb the local H-bond network, and have added a sentence to our discussion to highlight this caveat (lines 379-381). That being said, our conclusions do not solely rely on the T to A mutant. The phenotypes observed in our phosphomimetic mutant overlap with the TA mutant (increased sensitivity to MMS; defects in cell cycle re-entry after MMS treatment) (Figure 5). While the alanine mutation may not represent a purely ‘loss-of-phosphorylation’ state, our findings do implicate the importance of this residue in cell cycle control after DNA damage.

(3) It thus remains formally possible that pThr495 arises as an irrelevant side reaction due to activation of a kinase (with other relevant substrates).This dismal interpretation of the data would be dispelled somewhat if the stoichiometry of pThr495 were substantial, whereas very low stoichiometry of phosphorylation should leave one wary of the possibility that the surface-exposed Thr495 of ATP-bound Hsc70 is a physiologically irrelevant bystander target of a kinase activated in DNA-damaged cells.

We have included a Western blot in Figure 5 showing pHsp70 in our yeast samples. Here we can see low abundance of Hsp70 phosphorylation in untreated WT yeast, with a clear increase in MMS treated yeast. Additionally, as mentioned in a previous response, Figure 4e shows the accumulation of pHsp70 in human cells even after MMS removal, indicating it is not simply the byproduct of over-activation of the DNA damage response.

Unfortunately, the study does not quantify the stoichiometry of Hsp70 phosphorylation; detection relies on phospho-specific immunoblotting, leaving open the question of whether this modification occurs at physiologically significant levels. This worry is compounded by Figure 2a,f that suggests that phosphorylation occurs only under high-dose MMS or arsenite, raising concerns about physiological relevance.

We agree that we did not quantify absolute phosphorylation stoichiometry. While a precise measurement would be informative, our conclusions are based on regulated dynamics and functional perturbations rather than magnitude alone. Specifically, our pulse-chase (Figure 4e) shows that total Hsp70 levels remain constant while pHsp70 increases in a cell-cycle dependent manner following MMS removal. This indicates a regulated modification rather than a side-effect of kinase over-activation during acute stress. Additionally, perturbation of the homologous site produces cell-cycle phenotypes (Figure 5) in yeast, supporting functional relevance.

However, as mentioned in responses to Comment 3, our pulse-chase assay in Figure 4e indicates the stoichiometry of pHsp70 increases after MMS removal in a cell-cycle dependent manner. Furthermore, as discussed in response to Reviewer 1 Comment 8, Figure 4d highlights a technical limitation with regards to detection of pHsp70 by Western blotting. Namely, as pHsp70 accumulates in the nucleus, signal appears to be diluted by unmodified Hsp70 in the cytosol when whole-cell lysate is probed, thereby reducing detection capacity. It is therefore possible that less stringent doses do lead to phosphorylation, but due to the experiments being run in asynchronous cells and on whole cell lysate we failed to detect it.

**Reviewer #3 (Recommendations for the authors):**
Major Comments:(1) Figure 1e - Which antibody was used to probe this blot?

Thank you for catching this omission. This was stained with Coomassie. We have edited the figure legend to reflect this.

(2) Figure 1c- Do the authors have the data of the WT and T495E with DJA2?

The assay was performed with increasing concentrations of DJA2 for both constructs (from 0 µM to 4 µM) (lines 118-119, Figure 1c).

(3) Figure 2- The labeling of the right side of the immunoblots is missing.

We apologize for the confusion. The labeling is on the left. The lines on the right are intended to demarcate blots that came from the same membrane (for easier comparison of loading controls).

(4) Figure 2d- Does MMS treatment lead to a heat shock response?

We have not directly tested this. However, we do not see the massive upregulation of HSPs that would be expected from a heat shock response.

(5) Figure 4c and e - Total protein level of some of the phospho-proteins is missing.

We used housekeeping proteins as loading control. We do not have antibodies for all the non-phospho proteins. For those we have, blots not included in the publication do not show any marked discrepancies between the non-phospho form and the housekeeping proteins.

(6) Figure S1A- Although the authors suggest that the phosphorylation event is reversible, they have not integrated it into the final model in Figure 6.

In line 403 we postulate that dephosphorylation may permit client release. In the interest of clarity, we have now removed the model figure.

(7) Yeast genotype is missing.

We used W303a yeast (line 612).

(8) It is unclear which phosphatase inhibitor was used in their assay (Figure S1A).

We repeated the experiment with both Halt Phosphatase Inhibitor Cocktail (Thermo Scientific 78440) and Roche PhosStop (Roche 04906837001) (lines 524-525).

(9) Please add this most recent and up-to-date reference (PMID: 40976416) related to your study.

We have now added that reference

(10) Can the authors speculate on whether Hsp70- T495E is expected to primarily reside in the nucleus?

We have no data to indicate whether or not phosphorylation at T495 or a phosphomimetic mutation in this site would directly affect nuclear import or export. In cells expressing the *Legionella* kinase LegK4, pHsp70 exists in the cytoplasm,[1] indicating the phosphorylation in of itself does not force nuclear localization. We thus imagine that the nuclear localization seen in Figure 4d is more likely due to the location of the kinase rather than as a consequence of the phosphorylation. In an over-expression system or in the case of a genomic mutation, we believe the protein is most likely to exist in both the cytoplasm and in the nucleus, though we did not directly test this.

References

(1) Moss, S. M. et al. A *Legionella pneumophila* Kinase Phosphorylates the Hsp70 Chaperone Family to Inhibit Eukaryotic Protein Synthesis. Cell Host Microbe 25, 454-462.e6 (2019).

(2) Holt, L. J. et al. Global Analysis of Cdk1 Substrate Phosphorylation Sites Provides Insights into Evolution. Science 325, 1682–1686 (2009).

(3) Choi, S., Gamper, A. M., White, J. S. & Bakkenist, C. J. Inhibition of ATM kinase activity does not phenocopy ATM protein disruption. Cell Cycle 9, 4052–4057 (2010).

(4) Menolfi, D. & Zha, S. ATM, ATR and DNA-PKcs kinases—the lessons from the mouse models: inhibition ≠ deletion. Cell Biosci. 10, 8 (2020).

(5) Weiss, W. A., Taylor, S. S. & Shokat, K. M. Recognizing and exploiting differences between RNAi and small-molecule inhibitors. Nat. Chem. Biol. 3, 739–744 (2007).

(6) Kim, S.-T., Lim, D.-S., Canman, C. E. & Kastan, M. B. Substrate Specificities and Identification of Putative Substrates of ATM Kinase Family Members*. J. Biol. Chem. 274, 37538–37543 (1999).

(7) Jette, N. & Lees-Miller, S. P. The DNA-dependent protein kinase: A multifunctional protein kinase with roles in DNA double strand break repair and mitosis. Prog. Biophys. Mol. Biol. 117, 194–205 (2015).

(8) O’Neill, T. et al. Determination of Substrate Motifs for Human Chk1 and hCds1/Chk2 by the Oriented Peptide Library Approach*. J. Biol. Chem. 277, 16102–16115 (2002).

(9) Fulcher, L. J. & Sapkota, G. P. Functions and regulation of the serine/threonine protein kinase CK1 family: moving beyond promiscuity. Biochem. J. 477, 4603–4621 (2020).

(10) Craig, A. et al. Allosteric effects mediate CHK2 phosphorylation of the p53 transactivation domain. EMBO Rep. 4, 787–792 (2003).

(11) Xiao, W., Chow, B. L. & Rathgeber, L. The repair of DNA methylation damage in *Saccharomyces cerevisiae*. Curr. Genet. 30, 461–468 (1996).

(12) Memisoglu, A. & Samson, L. Base excision repair in yeast and mammals. Mutat. Res.Fundam. Mol. Mech. Mutagen. 451, 39–51 (2000).

(13) Srivastava, D. K. et al. Mammalian Abasic Site Base Excision Repair IDENTIFICATION OF THE REACTION SEQUENCE AND RATE-DETERMINING STEPS*. J. Biol. Chem. 273, 21203–21209 (1998).

(14) Preissler, S., Rato, C., Perera, L. A., Saudek, V. & Ron, D. FICD acts bifunctionally to AMPylate and de-AMPylate the endoplasmic reticulum chaperone BiP. Nat. Struct. Mol. Biol. 24, 23–29 (2017).

(15) Fontaine, S. N. et al. Isoform-selective Genetic Inhibition of Constitutive Cytosolic Hsp70 Activity Promotes Client Tau Degradation Using an Altered Co-chaperone Complement*. J. Biol. Chem. 290, 13115–13127 (2015).